# Genomic landscape of platinum resistant and sensitive testicular cancers

Chey Loveday [1,19], Kevin Litchfield[1,2,19], Paula Z. Proszek[3], Alex J. Cornish[1], Flavia Santo[3], Max Levy[1], Geoff Macintyre [4], Amy Holryod[1], Peter Broderick [1], Darshna Dudakia[1], Barbara Benton[1], Maise Al Bakir[2], Crispin Hiley[2], Emily Grist[5], Charles Swanton [2,6,7], Robert Huddart [8], Tom Powles[9], Simon Chowdhury[10], Janet Shipley[5,11], Simon O'Connor[3,12,13], James D. Brenton [4], Alison Reid[14], David Gonzalez de Castro [15], Richard S. Houlston [1,5] & Clare Turnbull[1,16,17,18✉]

While most testicular germ cell tumours (TGCTs) exhibit exquisite sensitivity to platinum chemotherapy, ~10% are platinum resistant. To gain insight into the underlying mechanisms, we undertake whole exome sequencing and copy number analysis in 40 tumours from 26 cases with platinum-resistant TGCT, and combine this with published genomic data on an additional 624 TGCTs. We integrate analyses for driver mutations, mutational burden, global, arm-level and focal copy number (CN) events, and SNV and CN signatures. Albeit preliminary and observational in nature, these analyses provide support for a possible mechanistic link between early driver mutations in *RAS* and *KIT* and the widespread copy number events by which TGCT is characterised.

[1] Division of Genetics & Epidemiology, The Institute of Cancer Research, London, UK. [2] Cancer Research UK Lung Cancer Centre of Excellence, University College London Cancer Institute, London, UK. [3] The Centre for Molecular Pathology, The Royal Marsden NHS Trust, Sutton, London, UK. [4] Cancer Research UK Cambridge Institute, University of Cambridge, Cambridge, UK. [5] Division of Molecular Pathology, The Institute of Cancer Research, London, UK. [6] Cancer Research UK Lung Cancer Centre of Excellence, UCL Cancer Institute, London, UK. [7] Translational Cancer Therapeutics Laboratory, UCL Cancer Institute, London, UK. [8] Academic Radiotherapy Unit, Institute of Cancer Research, London, UK. [9] Barts Cancer Institute, Queen Mary University, London, UK. [10] Department of Oncology, Guys and St Thomas' NHS Foundation Trust, London, UK. [11] Division of Cancer Therapeutics, The Institute of Cancer Research, London, UK. [12] Addenbrooke's Hospital, Cambridge, UK. [13] Department of Oncology, University of Cambridge, Cambridge, UK. [14] Academic Uro-oncology Unit, The Royal Marsden NHS Foundation Trust, Sutton, London, UK. [15] Centre for Cancer Research and Cell Biology, Queen's University Belfast, Belfast, UK. [16] William Harvey Research Institute, Queen Mary University, London, UK. [17] Guys and St Thomas' NHS Foundation Trust, Great Maze Pond, London, UK. [18] Public Health England, National Cancer Registration and Analysis Service, London, UK. [19]These authors contributed equally: Chey Loveday, Kevin Litchfield. ✉email: clare.turnbull@icr.ac.uk

Testicular germ cell tumours (TGCTs) are the most common cancer affecting young men, with peak incidence at age 30–36 years[1,2]. While cure rates for TGCTs are generally high due to their exquisite sensitivity to platinum-based chemotherapies, options are limited for the ~10% of patients who are platinum resistant, a group for whom the long-term survival rate remains poor[3,4]. Upfront identification of patients who are likely to be platinum resistant offers opportunities for tailoring management and leveraging targetable molecular dependencies for new treatments.

The major TGCT histologies are seminomas, which resemble undifferentiated primary germ cells, and non-seminomas, and show varying degrees of differentiation[5]. Approximately 5% of GCTs are diagnosed at extragonadal sites and display the same spectrum of histological types. Nonseminomas, and in particular extragonadal nonseminomas are associated with a lower rate of response to platinum[6,7].

TGCTs are characterised by hypertriploid to subtetraploid karyotypes featuring multiple aneuploidies. Gain of chromosome arm 12p is a near universal feature, along with elevated rates of reciprocal copy number changes. Frequent arm-level gains target chromosome 7,8,21,22 and X[8–10]. TGCTs have a low frequency of recurrent somatic mutations, with *KIT* and *KRAS* mutations most consistently reported as driver genes[9–16].

The molecular basis of platinum response in TGCT remains poorly understood. Defective homologous recombination and mitochondrial priming have both recently been proposed as possible mechanisms[12,17–19]. Increased frequency of *TP53* mutation and higher mutational burden has variously been reported in resistant tumours[13,14]. However, previous comparisons between resistant and sensitive tumours have provided inconsistent data, based on small sample series with failure to account for differences between series in histology, stage, site and sequencing platform[12–14,20,21].

Here, we present whole exome-sequencing (WES) on 40 tumours from 26 cases with platinum-resistant TGCT, the largest series to date, along with molecular inversion probe (MIP) analysis for copy number. We combine these data with published TGCT sequencing studies on 624 tumours from 605 cases, undertaking multifactorial logistic/linear regression analysis, adjusting for major clinical variables and other sources of bias. From this we present a comprehensive survey of the genomic landscape of TGCT to define the molecular hallmarks associated with response to platinum-based therapy.

## Results

**ICR2 series of platinum-resistant tumours**. We recruited a series of TGCT cases with platinum-resistant disease (ICR2 series), defined as showing progressive or viable disease following one or more completed regimens of platinum-based chemotherapy (Supplementary Table 1) and collected all available TGCT specimens for these cases. Following application of pre-sequencing and post-sequencing quality metrics, data for 22 primary and 18 metastatic tumours from 26 TGCT cases were included, comprising sequence data from 189,028 exons from 23,585 genes, and MIP analysis, targeting genome wide copy number aberrations (CNAs) in a subset of 33 tumours from 22 cases. We identified in total 1472 somatic small variants, comprising 1450 single nucleotide variants (SNVs) and 22 small insertion–deletion (indel) variants, of which 964 were nonsynonymous, along with 792 arm level and 1035 focal gain/losses.

**Additional TGCT series**. We combined ICR2 WES data with three previously published WES datasets to generate a combined WES series of 290 TGCTs from 267 cases including 48 (18%) with platinum-resistant disease (26 ICR2 + 22 DFCI). We also made use of mutation panel sequencing data for 261 cancer genes from two additional studies comprising 374 TGCTs from 364 cases (including 196 (54%) with platinum-resistant disease). In total, the full sample series from the six studies provided information on 664 tumours (444 primary, 220 metastatic) from 631 TGCT cases including 244 (39%) with platinum-resistant disease (Fig. 1 and Supplementary Fig. 1; Supplementary Tables 2, 3; Supplementary Data 1). For analyses requiring a single sample per TGCT case, a representative (index) sample was selected.

**Burden of small coding variants**. Across the combined WES series (290 TGCTs), a total of 4166 nonsynonymous variants and 1654 synonymous/noncoding somatic variants were detected, equating to a mean total mutational burden of 0.43 Mb (range 0–13.0 Mb) and a mean nonsynonymous mutational burden (TMB) of 0.33 Mb (range 0–9.4 Mb); results broadly similar to those previously documented[9–12] (Supplementary Data 2, 3; Supplementary Fig. 1a). In multifactorial adjusted analysis including adjustment for purity, TMB was significantly higher in platinum-resistant than in sensitive TGCTs ($p = 0.01$), with a mean increase of ~0.35 nonsynonymous mutations per megabase. This association remained significant when the analysis was confined to index primary tumours ($p = 0.003$), suggesting predictiveness in primary tumour for platinum resistance of TMB (Fig. 2). There was no significant association of TMB with histological subtype ($p > 0.1$ in all analyses, Supplementary Fig. 2). TMB was positively correlated with age at diagnosis, more prominently in nonseminomas (cor = 0.43, $p$-value = 0.0002), than seminomas (cor = 0.17, $p$-value = 0.17).

**Identification of driver genes**. To identify TGCT driver genes, we applied two complementary algorithms, OncoDriveFML[22] and MutSigCV[23] to index tumours in the combined WES series (267 tumours) and further explored their mutation frequency and distribution by integration of sequence data on 261 cancer genes from the full sample series (631 tumours). *KRAS* and *KIT* had the strongest evidence as driver genes from the combined analyses and, followed by *NRAS*, were the most frequently mutated genes (Fig. 3a and Supplementary Data 4 and 5; Supplementary Figs. 1a, 3). All three genes showed predominantly clonal mutation consistent with truncal events (Supplementary Data 6). We identified additional putative driver genes including the oncogenes *RAC1, PIK3CA, CBL,* and *CTNNB1* and tumour suppressor genes *KMT2C, CREBBP,* and *BCOR* (each $p < 0.05$ in either OncoDriveFML and/or MutSigCV analyses). Both *TP53* and *PTEN* were amongst the most frequently mutated genes in the full sample series. However, owing to their significantly lower mutational frequency in the exome analysis (*TP53* $p = 7.0 \times 10^{-8}$, *PTEN* $p = 9.4 \times 10^{-4}$ exome vs. panel data), these genes did not emerge as significant in the driver gene analysis (Supplementary Data 5).

**Clinical-pathological associations of driver genes**. Using multifactorial adjusted analysis, we identified a number of distinct clinical and pathological patterns of association for mutations in the (putative) driver genes (Fig. 3b and Supplementary Table 4). Mutation of *KIT* was associated with platinum sensitivity ($p = 0.0008$) whereas *TP53* mutation was associated with platinum resistance ($p = 0.03$). Association with seminoma was found for mutations in *KIT, KRAS* and *NRAS* and for putative oncogenic driver genes *CBL, RAC1, PIK3CA,* and *CTNNB1* (analysed jointly due to lower frequency), $p = 3.33 \times 10^{-10}$, $p = 1.41 \times 10^{-6}$, $p = 0.002$, $p = 0.009$, respectively. *TP53* mutation was associated with

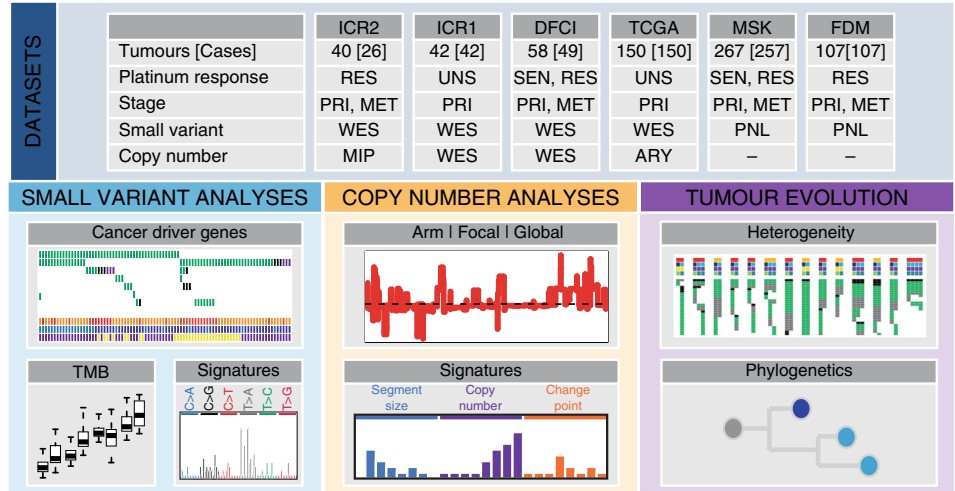

**Fig. 1 Study overview.** We combined data from six tumour sequencing studies of TGCT, performing small variant, copy number and tumour evolution analyses to define the molecular hallmarks of TGCT and their response to platinum-based therapies. RES: resistant, SEN: sensitive, UNS: unselected, PRI: primary, MET: metastasis, WES: whole exome sequencing, PNL: gene panel sequencing, MIP: molecular inversion probe, ARY: genotyping array, TMB: tumour mutation burden.

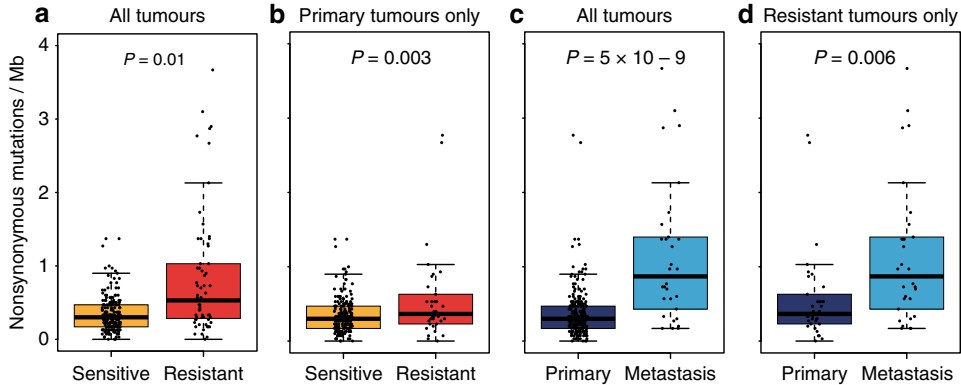

**Fig. 2 Tumour mutation burden in TGCT.** Box plot showing that tumour mutation burden (TMB) is higher in platinum-resistant tumours vs. platinum-sensitive/unselected tumours (**a**, **b**) and metastatic tumours vs. primary tumours (**c**, **d**). Analyses were performed on 269 tumours with exome sequencing data (from the ICR1, ICR2, DFCI, and TCGA sample series) after excluding those samples with missing histology and/or missing tumour purity estimate ($n = 20$), and one sample with an excessively high TMB (DFCI_C13_T1) ($n = 269$, **a**, **c**). Analyses were also restricted to a subset of tumours comprising either only primary tumours ($n = 239$, **b**) or tumours from cases with platinum resistant disease ($n = 64$, **d**). Boxes show the median ± 25–75th percentiles, whiskers show 1.5× interquartile range below and above the 25th and 75th percentiles, respectively. $p$ values are derived via two-sided multiple linear regression models adjusting for sample stage, histology, platinum response, dataset, and tumour purity.

nonseminoma ($p = 0.04$). Mutation of *KIT, KRAS, NRAS* and *TP53* were each associated with extragonadal disease ($p < 0.002$).

**Pathway analysis**. To explore pathways implicated in TGCT tumorigenesis, we organised known/putative TGCT driver genes into mutually exclusive gene sets, namely RAS/RAF, PI3K/MTOR and WNT/CTNNB1 signalling pathways, chromatin modification and DNA repair, along with other genes within those pathways carrying an oncogenic mutation. Mutation of *WNT/CTNNB1* pathway genes was associated with platinum-resistant ($p = 0.04$) and metastatic ($p = 0.03$) disease. The mutational profile was consistent with upregulation of WNT signalling, with activating mutations identified in *CTNNB1* ($n = 6$), and inactivating mutations in *APC* ($n = 3$), *FAT1* ($n = 3$), *AXIN1* ($n = 1$), and *GSK3B* ($n = 1$)) (Fig. 3b, c). Association was also identified for RAS/RAF with seminoma and extragonadal disease ($p = 2.9 \times 10^{-8}$ and $p = 4.7 \times 10^{-6}$, respectively) and PI3K/MTOR pathway with seminoma ($p = 0.02$, Fig. 3b).

**Copy number analysis**. Copy number analysis was performed on the 188/256 primary TGCTs from the complete WES series for which high quality copy number data, tumour purity > 40% and complete clinical data were available (Supplementary Fig. 1b, Supplementary Data 7, Supplementary Table 5). The majority of tumours had hyperploidy (80% had ploidy >2.5) and exhibited an elevated rate of arm-level amplifications (median number of arms/tumour with ≥1 allele amplified = 9).

The most frequently observed individual arm-level events included gain of 12p (95%) 21q (76%) and 7p (74%) and most frequent focal events were amplification of 12p13.32 (96%), 12p12.1 (93%), and 12p11.21 (92%) (Supplementary Data 7).

In multifactorial adjusted analysis, no significant differences were detected between platinum-sensitive and platinum-resistant primary tumours in the frequencies of global, individual arm or focal chromosomal aberrations (Fig. 4 and Supplementary Data 8, 9). Metastatic tumours showed an increased frequency of gains of 11p (30% vs. 3%, $p = 0.03$), 11q (30% vs. 3%, $p = 0.03$), 13q (30%

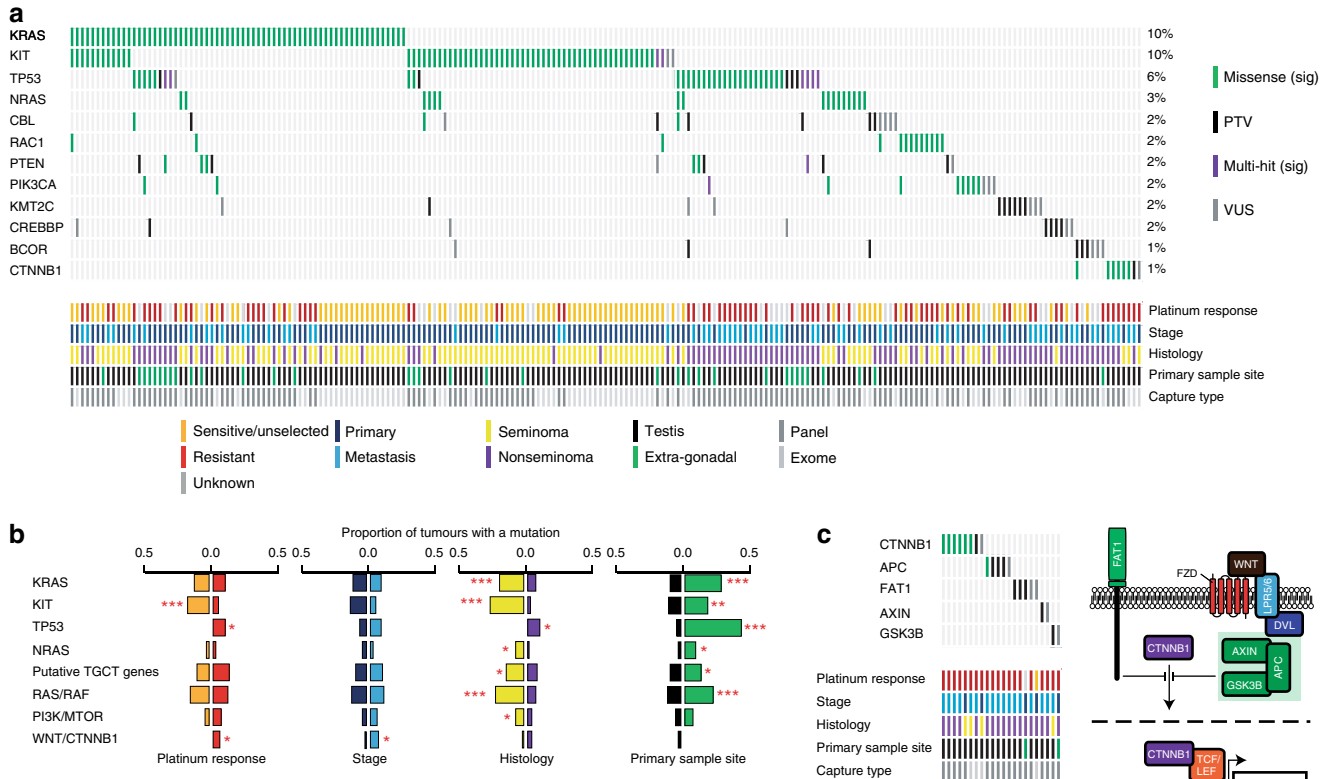

**Fig. 3 Mutational landscape of cancer driver genes in TGCT. a** Oncoplot showing the cancer driver genes most frequently mutated across 631 index TGCTs from the full sample series (ICR1, ICR2, DFCI, TCGA, MSK and FDM). Only TGCTs carrying a mutation in one of the displayed genes are shown ($n = 207$). **b** Oncogenic mutation frequencies in individual genes/gene sets across index TGCTs with the relevant available data, comparing: platinum-sensitive/unselected ($n = 284$, orange) vs. platinum-resistant ($n = 244$, red) tumours; primary ($n = 435$, dark blue) vs. metastatic ($n = 196$, light blue) tumours; seminomas ($n = 204$, yellow) vs. nonseminomas ($n = 417$, purple); and testicular ($n = 584$, black) vs. extragonadal ($n = 47$, green) primary sample site. Asterisks denote $p$ values derived from two-sided multivariable logistic regression, adjusting for platinum response, histology, stage, capture type, sample material and primary sample site.*$p < 0.05$, **$p < 0.01$, ***$p < 0.001$. Exact $p$ values are provided in Supplementary Table 4. Putative TGCT genes: *CBL*, *RAC1*, *PIK3CA*, *KMT2C*, *CREBBP*, *BCOR* and *CTNNB1*. **c** WNT/CTNNB1 pathway alterations are overrepresented in platinum-resistant metastatic TGCTs ($n = 22$) with the majority consistent with pathway activation. Pathway components that negatively regulate CTNNB1 are shown in green.

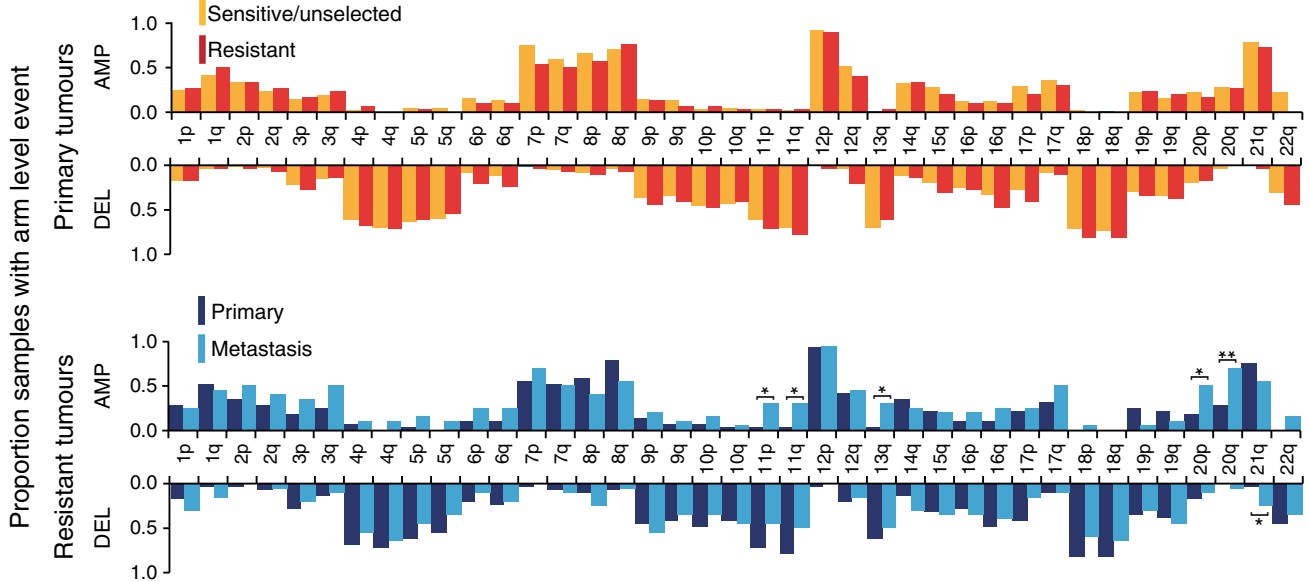

**Fig. 4 Autosomal arm level aneuploidy frequencies.** The frequency of whole arm autosomal aneuploidy events in 183 TGCTs with available copy number data and tumour purity ≥ 0.4 from series ICR1, ICR2, DFCI and TCGA, comparing platinum-sensitive/unselected (orange, $n = 134$) vs. platinum-resistant (red, $n = 29$) primary TGCT, and comparing primary (dark blue, $n = 29$) vs metastatic (light blue, $n = 20$) platinum resistant TGCT (only a single representative metastatic tumour from each patient with platinum resistant TGCT was included-see Supplementary Data 7). Asterisks denote $p$ values derived from two-sided multivariable logistic regression: *$p < 0.05$, **$p < 0.01$, Exact $p$ values are provided in Supplementary Data 8.

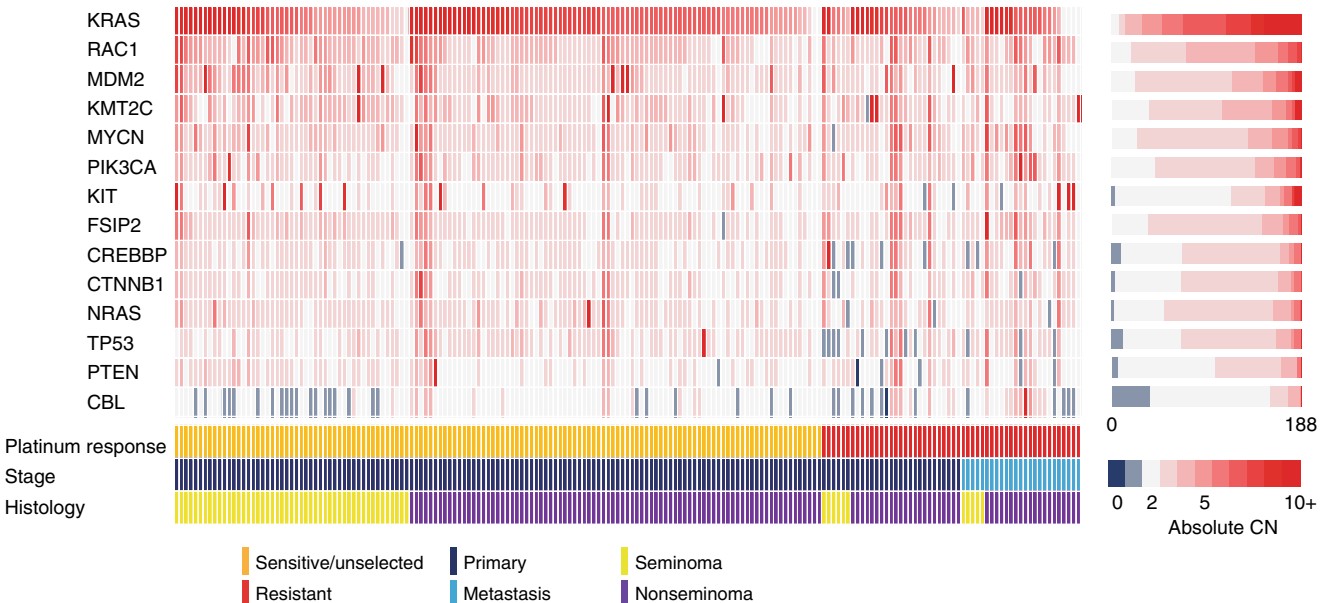

**Fig. 5 Absolute copy number of driver genes.** Plot showing the absolute copy number in 183 TGCTs with available copy number data and tumour purity ≥0.4 (ICR1, ICR2, DFCI and TCGA sample series) for fourteen genes including the putative/known TGCT driver genes (*KIT, KRAS, NRAS, RAC1, CBL, PIK3CA, CREBBP, KMT2C, CTNNB1*), *TP53* and *PTEN*, and three genes previously implicated as focally amplified in TGCT, namely *FSIP2, MDM2* and MYCN[9,10,13]. Genes are ordered vertically from most to least frequently demonstrating copy gain.

vs. 3%, *p* = 0.03), 20p (50% vs. 17%, *p* = 0.01) and 20q (70% vs. 28%, *p* = 0.004), loss of 21q (25% vs. 3%, *p* = 0.05), as well as focal amplification of 20q11.21 (65% vs. 34% *p* = 0.03) and focal deletion of 1p36.21 (25% vs. 33%, *p* = 0.01) (Fig. 4 and Supplementary Data 8 and 9).

We also assessed absolute copy number for each of the known and putative TGCT genes along with *MDM2, MYCN* and *FSIP* (which have previously been reported to be focally amplified in TGCT[9,10,13]) (Fig. 5 and Supplementary Table 6). *KRAS, RAC1* and *MDM2* were the genes most frequently affected by copy gain (5+ copies per tumour in 86%, 24% and 21%, respectively) whilst *CBL, TP53* and *CREBPP* were the genes most frequently affected by copy loss (0 or 1 copies in 20%, 5% and 4%, respectively). Multifactorial adjusted analysis did not identify any significant associations with platinum response for gain/loss of specific genes, individually or in combination (Supplementary Table 6)

**Mutational signatures**. We next examined signatures of mutational processes to gain insights into the molecular development of TGCT (Supplementary Fig. 1a). We analysed the 247 primary TGCTs with histolgy from the combined WES series for the 30 SNV mutational signatures defined by COSMIC[24] (Fig. 6 and Supplementary Data 10). SNV Signature 3 (SNV-Sig-3), indicative of failure of DNA double-strand break-repair by homologous recombination, was the predominant SNV signature in TGCTs, accounting for up to 40% of the mutational contribution (Fig. 6 and Supplementary Data 10). Markedly lower levels of SNV-Sig-3 were observed in platinum-resistant tumours. Conversely, SNV-Sig-1 was absent in platinum sensitive/unselected seminomas (Fig. 6 and Supplementary Data 10). Seminomas typically show global hypomethylation whilst nonseminomas exhibit increased methylation associated with increased differentiation from embryonal carcinoma through to teratoma. SNV-Sig-1 is thought to reflect spontaneous deamination of 5-methylcytosine. The absence of SNV-Sig1 observed in platinum sensitive/unselected seminomas is thus in fitting with existing hypotheses of

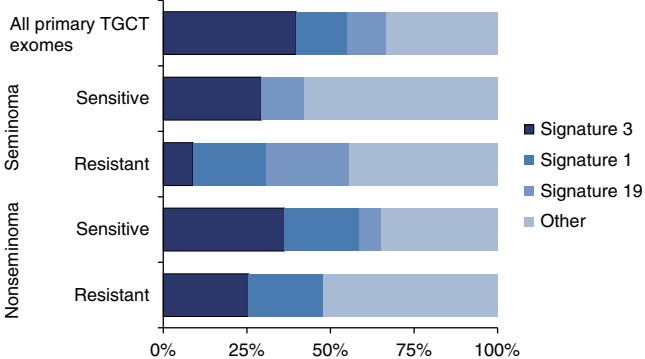

**Fig. 6 COSMIC mutational signatures.** COSMIC SNV mutation signatures identified in TGCT, organized by histology and platinum response. Analyses were performed on 247 primary tumours with exome sequencing data (ICR1, ICR2, DFCI and TCGA series) after excluding nine primary tumours with missing histology. Signatures were derived from somatic SNVs from primary tumours pooled together within each group as indicated. COSMIC signatures 1, 3 and 19 are displayed separately, whilst other represents the contribution from the remaining signatures combined.

association in cancer between differentiation, global DNA methylation status and response to chemotherapy[25,26].

We performed sample-level SNV signature analysis for in the 17 tumours carrying ≥50 mutations, of which 16 were platinum resistant and 12 were metastatic. In the 12 platinum resistant metastatic samples, SNV-Sig-3 was present in nine and predominant in five (Supplementary Data 10). The median value of SNV-Sig-3 in these 12 tumours was 0.13 (interquartile range: 0–0.37).

We next examined copy number signatures using non-negative matrix factorisation (NMF) as previously described[27]. We first applied the methods to fresh tissue samples of purity ≥0.4 with available array genotyping data (*n* = 105, TCGA, unselected TGCTs) (Supplementary Fig. 1b). Using random permutations of

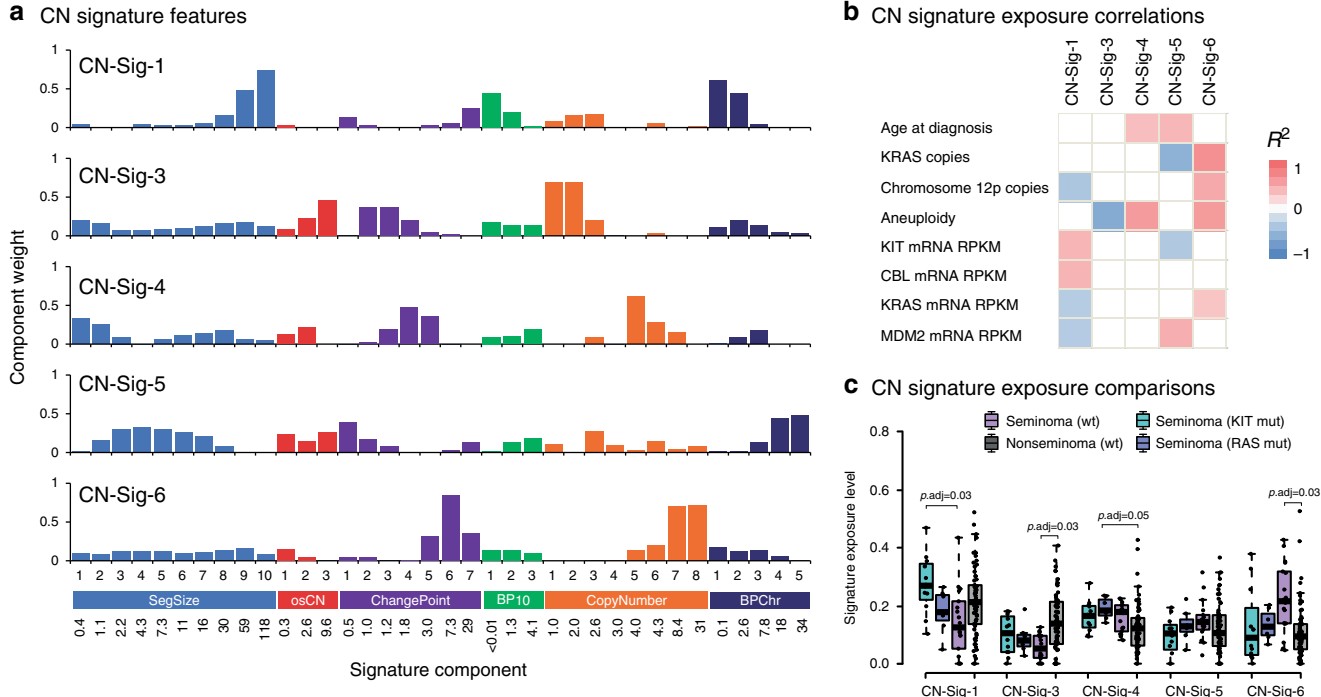

**Fig. 7 TGCT copy number signatures.** Five copy number signatures (CN-Sig-1, CN-Sig-3, CN-Sig-4, CN-Sig-5, CN-Sig-6) identified using TCGA SNP array data ($n = 105$). **a** Defining features of the CN signatures, showing each feature (SegSize, osCN, ChangePoint, BP10, CopyNumber and BPChr) split into 36 constituent components, as defined in ref. [27]. The mean value for each component is shown on the $x$ axis, with component weights shown on the $y$ axis. Features are defined as follows: SegSize, segment size (Mb); ocCN, region length with neighbouring oscillating copy number segments (Mb); ChangePoint, difference in copy number between neighbouring segments; BP10, number of break points (10 Mb$^{-1}$), CopyNumber, absolute copy number of segment; BPChr, breakpoints per chromosome arm. **b** CN signature exposure correlations with clinical and molecular features. Two-sided Pearson's correlations for association of $q < 0.05$ are illustrated. Age, aneuploidy and chromosome 12p copies: analysed across histologies. RAS pathway-related features: analysed in seminomas only. **c** Box plots showing CN signature exposure comparisons for *KIT* ($n = 12$) and *RAS* ($n = 7$, *NRAS/KRAS*) mutated seminomas, wildtype seminomas ($n = 18$) and nonseminomas ($n = 66$). Boxes show the median ± 25–75th percentiles, whiskers show 1.5× interquartile range below and above the 25th and 75th percentiles, respectively. $p$ values are derived from two-sided Wilcoxon Rank Sum tests adjusted for multiple comparisons via the Benjamini-Hochberg method.

the data and four model selection measures, we found the optimal number of signatures in the TGCT dataset to be five, and each signature was present at >5% in 74–91% of TGCTs and shared highly similar component weights to those previously reported ($p < 5.9 \times 10^{-9}$, $r^2 \geq 0.8$) (Fig. 7 and Supplementary Data 11; Supplementary Tables 7, 8; Supplementary Fig. 4)[27].

CN signature 1 (CN-Sig-1), characterised by large segment sizes and low breakpoint frequency (0-2 breakpoints per chromosome arm), was the predominant signature and was higher in seminomas positive for mutations in *KIT* ($p = 0.004$) and *RAS* mutations (*KRAS* and *NRAS* combined, $p = 0.07$) (Fig. 7c), recapitulating the association of mutations in the RAS/MAPK signalling pathway with CN-Sig-1 reported in serous ovarian carcinomas[27]. The next most significant signature exposure was CN-Sig-3, characterised by evenly distributed chromosome breaks and diploid-to-single copy changes (i.e., loss of heterozygosity), which in ovarian carcinoma was associated with defective homologous recombination and somatic/germline *BRCA1/2* mutation[27].

Both CN-Sig-4 and CN-Sig-6 have previously been associated with failure of cell cycle control, and were shown to be associated with seminoma histology ($p < 0.007$; Supplementary Fig. 4c). Notably CN-Sig-6, which reflects high copy number states from focal amplification, was particularly high in seminomas with (i) increased 12p copy number, (ii) *KRAS* copy number and/or, (iii) *KRAS* mRNA expression (without *KIT/KRAS/NRAS* mutations) (Fig. 5b and Supplementary Table 8). Also identified was CN-Sig-5, which is characterised by subclonal copy number changes and a

large number of breakpoints per chromosome arm; CN-Sig-5 is proposed to reflect chromothriptic-like events and/or increased subclonal diversity. We found CN-Sig-5 to be associated with MDM2 expression in nonseminomas ($p = 0.00008$) and to be correlated with age at diagnosis in seminomas ($p = 0.01$).

We next applied CN signature analysis to a second series of cases in which copy number had been called from fresh-tissue-derived WES data ($n = 19$, ICR1, unselected TGCTs). Detection of the same five CN signatures was reproduced, along with the associations by histological subtype (Supplementary Table 9; Supplementary Fig. 4c). However, detection of the CN signatures was less delineated in FFPE-derived sample series ICR2 and DFCI, preventing analysis of potential differences in CN signature exposure between platinum-sensitive vs platinum-resistant tumours (Supplementary Table 9).

**Longitudinal analysis of platinum-resistant tumours.** Restricting analysis to those platinum-resistant cases for which serial samples were available, we analysed 39 tumours from 17 individuals (ICR2, DFCI).

Overall, for the 1126 unique SNVs identified, there was wide variability between patients in their inter-tumour mutational concordance (Jaccard index ranged from 2 to 92) (Supplementary Table 10; Supplementary Fig. 5). We performed gene function enrichment analysis on nonsynonymous mutations grouped by whether they were (i) truncal events (i.e., shared between primary and metastatic tumours), (ii) private amongst pre-treatment

primary tumours or (iii) private amongst post-treatment metastatic tumours. Notable gene sets enriched in truncal events include those associated with DNA binding (GO-MF:0003677; $p = 0.003$), nucleobase metabolism (GO-BP:0019219; $p = 0.05$), and base excision repair (KEGG:03410; $p = 0.03$) (Supplementary Data 12). No gene set enrichments were found for nonsynonymous variants private to primary tumours or for silent mutations in any group.

We performed phylogenetic analyses on large-scale CNVs identified in ICR2 primary and metastatic samples using MEDICC (Minimum Event Distance for Intra-tumour Copy number Comparisons)[28,29]. This revealed evidence for branched evolution in the vast majority of cases (Supplementary Figure 6). In the two cases where data was available for multiple metastatic tumours, the metastatic samples descended from a single common metastatic precursor. Metastatic samples showed continued acquisition of copy number aberrations, and a linear relationship was observed between the number of CN events and number of months since initial diagnosis when looking across all metastatic samples (Supplementary Table 11; Supplementary Fig. 7; $r^2 = 0.70$; $p = 0.02$).

## Discussion

Platinum-resistant TGCT remains an unsolved clinical problem[30]. We have assembled the largest WES series to date of platinum-resistant tumours ($n = 40$, ICR2), with which we have integrated five additional series to present data on 664 tumours, far and away the largest TGCT series analysed to date. In summary, we demonstrate via multifactorial analysis adjusted for clinical and technical confounding factors, that there is significant difference between platinum-sensitive and platinum-resistant tumours in frequency of (i) *KIT* mutation, (ii) *TP53* mutation, (iii) mutations in WNT/CTNNB1 signalling genes, and (iv) mutational burden of nonsynonymous small variants. Furthermore, we delineate the prominence in TGCT of mutational signatures indicative of HRD (SV-Sig-3, CN-Sig-3) and RAS-related structural aberration (CN-Sig-1). Integrating the full data from platinum sensitive and platinum resistant tumours, we can distil into seven molecular hallmarks our current understanding of TGC tumorigenesis and platinum response.

Firstly, TGCTs are characterised by a TMB substantially lower than that of other adult solid tumours and more consistent with paediatric embryonal tumours. Consistent with the proposed embryonal aetiology of TGCT, TMB is correlated with age of tumour diagnosis. Previous reports have highlighted a higher TMB observed in post-chemotherapy metastatic samples as compared to treatment-naïve (subsequently platinum-sensitive) TGCTs: the direct effects of chemotherapy on TMB cannot be partitioned here[13,14]. We demonstrate that in primary TGCT, TMB is significantly higher in platinum-resistant tumours, independent of subsequent chemotherapy treatment. This strongly suggests that treatment resistance may be a consequence of other mutational processes that provide a larger substrate of genetic variation that increases the chances of evolutionary rescue. Intriguingly, we found enrichment amongst truncal events of mutations in genes involved in inter-strand crosslink resolution. TMB itself is widely accepted to be a barometer of neoantigen load and tumour immunogenicity. Consistent with this, there are several case reports in platinum-resistant TGCTs of response to checkpoint inhibitors (reviewed in ref. [31]); nevertheless, detailed correlation against expression of specific antigens has yet to be performed.

Secondly, the small variant landscape of TGCT is dominated by mutation of *KIT* and *RAS/RAF*-pathway genes. Through adjusted analysis, we delineate independent association with seminoma histology for mutations in *KIT*, *KRAS*, *NRAS* and *Pi3K/MTOR* pathway gene sets and, independent of histology, association of *KIT* mutation with platinum sensitivity. This enrichment in seminomas is consistent with mutation of *KIT* and its downstream effectors occurring at an early stage of TGC tumorigenesis and blocking differentiation from the inherently platinum-sensitive PGC-like seminomatous state[10].

Small migratory populations of primordial germ cells (PGCs) first arise in the early embryo and pass through various embryonic tissues on their way to the nascent gonad[32]. Extra-gonadal GCTs, located along the PGC migration path, likely reflect disruption to this process. In adjusted analysis, we observed enrichment in extragonadal GCTs compared to gonadal GCTs for all TGCT driver genes and pathways analysed (*KRAS, KIT, NRAS*, seven putative driver genes, other RAS/RAF pathway genes), providing support for the hypotheses that TGCT driver mutations (i) typically arise prior to PGC separation and (ii) increase the probability that a PGC will become misplaced. Model organisms with mutations in genes orthologous to *KIT* and *TP53* display an increase in PGCs outside the gonad, supporting the importance of correct migration and survival of PGCs for these genes[33].

Indeed, the third hallmark of TGCT is the overall low frequency of somatic *TP53* mutation (6.7%), with mutations identified predominantly in non-seminomatous TGCTs and extragonadal TGCTs (both parameters strongly associated with platinum resistance). Following adjustment for the confounding influences of site, histology and sequencing platform, the independent association of *TP53* mutation with platinum resistance is marginal ($p = 0.03$), a result that contrasts with previous unadjusted analyses. Furthermore, there was no association ($p = 0.81$) with platinum resistance of amplification of *MDM2*, a *TP53* antagonist amplified in a sizeable proportion (20%) of TGCTs. We identified association between *TP53* mutation and panel sequencing methodology, residual after adjustment for potentially confounding parameters. *TP53* variant allele frequencies were typically high, and coverage of *TP53* was sufficiently adequate in the exomes, to rule out differential detection of subclonal *TP53* mutations between the two methods. It is possible the observed association may be due to confounding factors unaccounted for in these analyses, for example systematic differences in the clinical make-up of the constituent sample series.

The fourth hallmark of TGCT is aberration of the WNT/CTNNB1 signalling pathway, with enrichment for somatic WNT/CTNNB1 pathway mutations detected in both platinum-resistant and metastatic TGCTs. Previous immunohistochemical and expression analyses have exhibited upregulation of WNT/CTNNB1 signalling in TGCTs, particularly nonseminomas[10,34,35]. The WNT/CTNNB1 pathway plays an important role in multiple developmental processes and stemness and is upregulated in many other cancer types.

Fifthly, TGCTs are characterised by dramatic and widespread structural aberration. Early nondisjunction results in whole genome doubling and a tetraploid precursor cell, followed by gain and loss of multiple large chromosomal regions, most frequently isochromosome 12p. Like HGSOC, the extreme genomic complexity of TGCT has hampered mechanistic understanding of the aetiology/consequences of CN aberrations. Using CN signature analysis, we have been able to relate patterns of CN aberration to potentially causative molecular phenomena (Fig. 7).

For example, the high frequency of large but sub-whole arm exchanges in TGCTs is reflected in a high level of CN-Sig-1. We demonstrate clear association in seminomas of CN-Sig-1 with *KRAS/NRAS/KIT*-mutation, a phenomenon also observed in HGSOC[27]. Aberrant RAS signalling promotes genomic instability due to aberrant cell cycle checkpoint control leading to

chromosomal missegregation[36,37]. In HGSOC, CN-Sig-1 is directly correlated with telomere shortening and frequency of amplification-associated fold-back inversion events, pointing to the underlying mechanism as breakage-fusion-bridge[27]. In keeping with these observations, telomeric shortening is a consistent feature of *KIT/RAS* mutated seminomas[38].

Meanwhile, compared to *KIT/RAS* mutated tumours, wildtype seminomas exhibit strong exposure of CN-Sig-6 (indicating focal amplification with high copy number states). Rather than a mere passenger effect, the focal amplification and upregulation of *KRAS* in these wildtype seminomas may instead reflect an active alternative mechanism by which RAS signalling has been increased.

The sixth hallmark of TGCT is that progression and metastasis in resistant tumours are characterised by evolving aberration of copy number. Supported anecdotally via longitudinal phylogenetic analysis in select serially sampled tumours ($n = 5$)[12], we have now demonstrated this via systematic comparison across the series of primary and metastatic tumours. In particular, gain of chromosome arm 20q and focal amplification of 20q11.21 are each present in 65–70% of metastatic TGCT, representing statistically significantly elevated rates compared to primary resistant tumours. Recurrent gain of 20q has been observed across numerous cancer types and metastatic behaviour is associated with amplification of different sub regions of 20q[39–43]. Gain of 20q11.21 is very frequent in human embryonic stem cell (hESC) lines, human embryonal carcinoma cell lines and some teratocarcinomas[44], and may confer protection against apoptosis as a result of BCL2L1 overexpression[45,46], as well playing a role in malignant transformation[45].

Finally, homologous recombination deficiency (HRD) would appear to be a universal feature of TGCT, as exemplified by our observation of marked predominance of SNV-Sig-3 along with prominence of CN-Sig-3. Whilst we and others have found a low frequency of coding mutations in genes involved in homologous recombination repair, epigenetic silencing is a plausible underlying mechanism as a high frequency of inactivation by methylation of *BRCA1* and *RAD51C* has been reported in TGCTs[10,17]. Whilst signature assignation at cohort level indicates predominance of SNV-Sig-3 with inverse association with platinum resistance, robust insight into the aetiological contribution of HRD to TGCT and platinum response will require individual sample level analysis with de novo signature extraction, assiduous signature fitting and adjustment for confounding covariates[47]. On account of the low frequency of somatic mutation in TGCTs, whole genome rather than exome analysis will be required to further explore our preliminary observations.

Limitations of our study include the modest size of the ICR2 series (40 samples, 26 cases), resultant from the overall low frequency of TGCTs compounded by the small platinum-resistant proportion therein. Furthermore, as platinum response is only manifest downstream of diagnosis, only routine formalin-processed clinical specimens were available pre-treatment. For these same logistical reasons, very limited additional WES data were available for resistant TGCTs (20 cases) with the majority of molecular data from resistant TGCTs coming from routine clinical diagnostic gene panels. Thus, beyond comparison of frequency of specific small variants, the pan-genomic comparative analysis of resistant versus sensitive disease remains limited by power and sample preparation. Whilst we undertook multifactorial adjusted analysis to mitigate against the confounding influence of variability in constituency of cases across the six studies (histology, site, stage, specific study, sample preparation, and capture-type), integrated analyses across studies was compromised by heterogeneity in molecular testing platforms, in particular for analyses of copy number. The significant associations observed may reflect causal relationship but may also arise due to the influences of chance, bias, confounding or reverse causality.

In conclusion, we have presented a comprehensive molecular description of platinum-resistant and platinum-sensitive TGCTs based on assembly of the largest integrated genomic dataset to date. We demonstrate that platinum resistance cannot be accounted for by a single molecular entity but is likely to involve multiple mutational processes. To better understand these processes, further large series of serial fresh-processed platinum-resistant (and sensitive) TGCT samples are required with aligned longitudinal clinical data. Whilst whole genome sequencing data will be invaluable for additional exploration of the clinical and molecular correlates of platinum response and SNV and CN signatures, more comprehensive aetiological insight will likely require epigenetic, transcriptomic and proteomic profiling.

## Methods

**Cases and samples.** The ICR2 TGCT case series was ascertained and consented via the UK Genetics of Testicular Cancer Study, which has been reviewed by the Royal Marsden Hospital Committee for Clinical Research, Cancer Research UK programmes committee and received ethical approval from the South West MREC (research and ethics committee (06/MRE06/41). All patients provided written informed consent. Patients already consented within this study were selected for inclusion in the ICR2 series if platinum resistance was reported by the referring clinician, that is after one or more complete regimens of platinum-based chemotherapy there was progressive disease (relapse or incomplete response), or viable (non-teratomatous) disease in a post-chemotherapy surgical sample. All documented tissue samples for these patients were sought via contact with the respective pathology department. Stored formalin fixed paraffin embedded tumour tissue samples (slides/blocks) were sent to our centre and material was retrieved by a trained pathologist by macrodissection guided by hematoxylin and eosin-stained slides. Tumour DNA was extracted using Qiagen's QIAAMP® DNA FFPE tissue kit (Qiagen, Hilden, Germany) following manufacturers protocols. Tumour DNA samples were quantified and qualified using Qubit technology (Invitrogen, Carlsbad, CA, USA) and the 2200 TapeStation System (Agilent, Santa Clara, CA, USA). Matched normal DNA was obtained from lymphocytes using standard techniques. We affirm compliance with all relevant ethical regulations for work with human participants.

Other sample series were recruited as previously described[9,10,12–14,48].

**WES sample processing (ICR2).** DNA libraries for ICR2 tumour-normal pairs were prepared from 200 to 400 ng of DNA (depending on quality), using the KAPA HyperPlus Kit (Roche, Basel, Switzerland) with seven PCR cycles following standardized protocols as per manufacturer guidelines. Exome capture (4-plex) was performed using the SeqCap EZ HGSC VCRome kit (Roche, Basel, Switzerland) following standardized protocols as per manufacturer guidelines. Samples underwent paired-end sequencing using the Ilumina HiSeq2500 platform (Illumina, San Diego, CA, USA) with a 100-bp read length. FASTQ files were generated using Illumina CASAVA software (v.1.8.1, Illumina).

**WES alignment and variant calling (ICR2 and DFCI).** The ICR2 and DFCI whole exome sequencing data were processed through a common informatics pipeline. Sequence alignment was performed following GATK best practices. Raw unmapped reads were aligned to build 37 of the human genome using BWA-MEM (v0.7.12), and the resulting SAM files were converted to BAM files, and sorted and indexed using SAMtools (v1.3). Picard Tools (v1.94) was used to merge BAM files and to mark duplicates. Indel realignment and base quality were performed using IndelRealigner and PrintReads, respectively (GATKv3.6). Somatic variant calling was performed using Strelka (v1.0.14; for SNVs and indels), Mutect (v1.1.7; for SNVs) and Mutect2 (GATKv3.6; for indels). A panel of normals was generated using germline (normal) samples for use with Mutect and Mutect2. Variants were retained for analysis if called by ≥2 calling algorithms. Artefacts introduced by DNA oxidation during sequencing of FFPE were identified using the FoXoG algorithm[49] and removed. FoxoG ensured variants were supported by a minimum of one alternative read in each strand direction, a mean Phred base quality score of 26, mean mapping quality greater than or equal to 50 and an alignability site score of 1.0.

**WES quality control (ICR2 and DFCI).** The following filters were applied to remove low quality samples: percentage of bases at 15× <50% (DepthOfCoverage, GATKv3.6); samples with cross-individual contamination fraction >5% (ContEst, GATK v3.6). NGSCheckMate (v1.0) was used to confirm that both samples from each tumour-normal pair, and serial tumour samples for a given case, were from the same individual[50]. Mean target coverage across the cohorts for tumours was

130× and for normal samples was 97×. After the removal of two samples with ContEst scores of >5%, the mean ContEst value was 0.52%.

**WES SNV calls from other datasets (ICR1 and TCGA)**. ICR1 SNV calls were generated as previously described[9]. TCGA SNV calls were obtained from the BROAD Firehose browser[51].

**WES SNV hard filtering (ICR1, ICR2, DFCI and TCGA)**. Additional hard filters were applied to the pooled set of SNV calls from the exomic datasets to remove variant calls likely to be artefacts. SNVs were required to have five or more supporting reads, indels were required to have ten or more supporting reads, and both were required to have a variant allele frequency (VAF) of 0.05 or greater. Variant calls were rescued if (a) they overlapped with known driver mutations documented in OncoKB or (b) if they overlapped with a COSMIC alterations with a count of at least $n = 50$. A variant was also rescued if it passed these filters in a second tumour from the same individual. Any individual gene or variant with a mutation frequency greater than or equal to 10% within a single dataset whilst simultaneously having a frequency of less than 1% within each of the remaining datasets were also removed on the basis that they were likely an artefact specific to that dataset. Analyses of the combined WES series were restricted to variants located at the intersect of the different captures (comprising ~30 Mb of the genome), as determined using the intersect tool from bedtools (v2.25.0).

**SNV calls from panel data (MSK, FDM)**. SNV data were incorporated from a further two sources: (i) 278–315 genes analysed by Foundation Medicine, 107 cases with platinum-resistant TGCT, as published[14] and (ii) clinical sequencing data on 341–410 genes from 267 cases from Memorial Sloan Kettering patients (MSK-IMPACT), as published[13,48]. Variants with ambiguous annotation in the Foundation Medicine dataset were removed from all analyses.

**Identifying likely TGCT driver genes**. WES data: MutSigCV was employed to identify genes from the combined WES series that are mutated more often than one would expect by chance. This method corrects for variation by employing patient-specific mutation frequencies and mutation spectra, and gene specific mutation rates that incorporates covariate factors such as replication timing and expression levels[23]. OncodriveFML was employed to look for evidence of positive selection. This method works by estimating the accumulated function impact bias of somatic mutations based on a local simulation of the mutational processes affecting it[22]. Analyses were performed on a single representative (index) sample from each case to avoid upward bias of results.

WES+ panel data: The total number of mutations identified across each of 261 cancer-associated genes common to all six WES and panel captures was calculated. All somatic changes were annotated at variant-level as being of likely functional significance if they were a known cancer driver alteration, as defined by their presence in one three databases (i) OncoKB, a precision oncology knowledge base hand curated by experts as described in ref. [52], (ii) Cancer Hotspots, a set of significantly recurrently mutated residues identified by the algorithm described in refs. [53,54] (iii) 3Dhotspot, a set of mutations that show evidence of clustering in 3D protein structures identified by the algorithm described in ref. [55].

Genes were categorised with respect to their driver status in TGCT: Definitive, gene passing at exome wide significance ($q < 0.05$) via both MutSigCV and OncoDriveFML; Likely, gene passing at exome wide significance ($q < 0.05$) in MutSigCV or OncoDriveFM; Putative, gene (a) showing nominal statistically significant result in at least one algorithm ($p < 0.05$), (b) residing within the top 5th centile of genes recurrently mutated by known cancer driver variants, (c) comprising a predominance (>50%) of previously described driver variants.

***KIT*, *KRAS* and *NRAS* clonal status**. We estimated the clonal status of mutations in *KIT*, *KRAS* and *NRAS* as previously described[56]. We used variant allele fraction, ploidy and purity values to calculate the multiplicity of each variant, that is the number of alleles within a given tumour that carry a particular mutation, using >0.8 as likely indicative of a mutation being clonal[56].

**Mutation status association analyses**. We performed multivariable logistic regression (i.e., multifactorial adjusted analysis) to examine the relationship between gene mutation status and clinical characteristics including histology (seminoma/nonseminoma), stage (primary/metastasis), tumour site (testis/extragonadal), and platinum response (sensitive/unselected or resistant) whilst adjusting for potential technical confounders including capture type (exome/panel) and sample material type (fresh frozen/FFPE). Stepwise iterative analyses were performed, whereby the variable with the least significant $p$ value was removed from the model, and this was performed until only statistically significant variables remained. When a model contained a variable of interest with borderline significance, we also performed likelihood ratio testing to evaluate whether inclusion of the variable offered improved prediction of the data. Tests with a significant $p$-value indicate that the reduced model is not a better fit of the data (i.e., that the borderline associated variant should be included in the model). Logistic regression was performed using the glm() function (binomial family with logit link) and

likelihood ratio testing was performed using the anova() function (with a chi-squared test), via the R package stats (v3.5.1). These and subsequent R packages were used in R (v3.5.1) via RStudio (v1.1.463).

**Tumour mutation burden analyses**. Tumour mutation burden was calculated on a per sample basis as the number of mutations divided by the number of bases within the intersected BED file, and given as the number of mutations per megabase. Mutational burden was calculated both across all SNVs and by limiting analysis to nonsynonymous mutations. Mutational burden comparisons were performed on nonsynonymous mutation burden values using multiple linear regression, with nonsynonymous mutational burden as the continuous outcome and multiple categorical predictors, allowing us to assess the independence of relationships whilst simultaneously adjusting for potential confounders. Categorical predictors included in the model were dataset (ICR1, ICR2, DFCI, TCGA), sample site (primary, metastatic), platinum response (sensitive/unselected, resistant), histology (seminoma, nonseminoma), and whether multiple samples from a single case were included (yes, no). Age at diagnosis, a potential cofounder, was not included in the regression model due to substantial missingness across the datasets. Analyses were performed in (a) all tumours, (b) primary tumours only and (c) tumours from platinum resistant cases only. A single sample with an excessively high mutational burden compared to the rest of the cases was excluded from these analyses (DFCI_C13_T1). Analysis was performed using the lm() function from the R package stats (v3.5.1). Supplementary to these analyses, we used an exact, distribution free method using the lmp() function from the R package lmPerm (v2.1.0), which uses permutation instead of normal theory.

**Pathways analysis**. Genes with oncogenic mutations were organised into signalling pathways or cellular functions. Genes were assigned to pathways based on the results of functional experiments in the primary literature. We focused on those gene sets implicated by known/putative TGCT driver genes and comprised: RAS/RAF signalling (*KRAS*, *NRAS*, *BRAF*, *NF1*); PI3K/MTOR signalling (*PIK3CA*, *PIK3CB*, *PIK3R1*, *PIK3R2*, *MTOR*, *RICTOR*, *AKT1*, *PTEN*); WNT/CTNNB1 signalling (*CTNNB1*, *APC*, *AXIN1*, *GSK3B*, *FAT1*); DNA repair (*BRCA1*, *BRCA2*, *ATM*, *CHEK2*, *PALB2*, *POLE*, *TOP1*, *BARD1*, *PMS2*, *MUTYH*, *BAP1*, *CDK12*) and chromatin modification (*CREBBP*, *DNMT3A*, *NSD1*, *EP300*, *KMT2A*, *SMARCB1*, *SMARCA4*, *SETD2*, *KMT2C*, *KDM6A*, *ASXL1*, *ARID1A*, *ARID1B*, *ARID2*, *PBRM1*).

**COSMIC SNV signature analyses**. We used non-negative matrix factorisation employing the R package deconstructSigs (v1.8.0) to decompose from whole exome SNV calls the 30 signatures of the COSMIC mutational signature matrix[57]. TGCT tumours were analysed jointly, grouped at various levels by stage, histology, platinum response and dataset. Tumours were analysed at an individual level only where total SNV count ≥ 50. To compare the distribution of COSMIC signatures in TGCT to 32 other cancer types within the TCGA cancer project we used mSignatureDB[58].

**Somatic copy number calling (ICR2)**. CNV calls for the ICR2 series for constitutional and tumour-derived DNA were produced using the OncoSCAN CNV Plus Assay (Applied Biosystems, Waltham, MA, USA), which uses molecular inversion probe (MIP) technology with a genome-wide resolution of 300 kb and is optimised for use with highly degraded FFPE DNA samples[59]. Assays were prepared as per the manufacturer's recommendations. Arrays were stained and washed using the GeneChip® Fluidics Station and loaded onto a GeneChip® Scanner (Affymetrix, Santa Clara, CA) where fluorescence intensity was scanned to generate array images (DAT files). Array fluorescence intensity data (CEL) files were generated and used to produce OSCHP-TuScan files with the OncoScan® Console software (v1.3) using the FFPE Analysis including Matched Normal workflow and the appropriate annotation files.

**Somatic copy number calling (TCGA, ICR1, DFCI)**. TCGA SNP 6.0 array data were obtained in the form of CEL files the GDC legacy archive at https://portal.gdc.cancer.gov/legacy-archive/search/f. PENNCNV-Affy (v1.0.3) was used to generate LogR and BAF values following the recommended user guide steps 1.1, 1.2 and 1.4. Raw, non-integer allele-specific copy number calls were retained and allele-specific copy number analysis, ploidy and purity estimates of tumour samples were obtained using ASCAT (v2.1.1) using the R script ASCAT_fromCELfiles.R at GitHub. Allele specific somatic copy number calling, ploidy and purity estimates were obtained from WES data for the ICR1 and DFCI datasets as previously described[12]. The X chromosome was excluded from all analyses.

**Analyses for aneuploidy**. The fraction of the genome affected by aneuploidy was calculated by dividing the sum of the total number of bases within segments showing copy number changes (relative to ploidy rounded to nearest integer) divided by the total number of bases across all segments. Arm level events were called from segmented log2 copy ratio data using GISTIC (v2.0)[60] via GenePattern[61]. Arm level and focal analyses were performed using the default settings. Arm level and focal events were called as present if > absolute value of 0.1. Arm

level aneuploidy score was calculated by summing the number of autosomal arm level events within each tumour sample to generate a score between 0 and 39, reflecting events covering both the long and short arms of the non-acrocentric chromosomes and the long arms only of the acrocentric chromosomes (13, 14, 15, 21 and 22). Comparisons were performed on fraction of genome aneuploid and arm level aneuploidy scores, respectively, using multiple linear regression, with the aneuploidy measure as the continuous outcome and categorical predictors comprising dataset (ICR1, ICR2, DFCI, TCGA), sample site (primary, metastatic), platinum response (sensitive/unselected, resistant) and histology (seminoma, nonseminoma). Analysis was performed using the lm() function from the R package stats (v3.5.1). Supplementary to these analyses, we used an exact, distribution free method using the lmp() function from the R package lmPerm (v2.1.0) to confirm associations independently of the underlying data distribution. Fisher's exact test was performed in order to compare the number of events between (a) primary TGCT tumours with platinum-sensitive vs. platinum-resistant disease and (b) between primary and metastatic tumours from patients with platinum resistant disease. Associations at a nominally statistically significant $p$ value of <0.05 were further analysed by multivariable logistic regression. These analyses were performed in a subset of 183 tumours with purity estimates ≥ 0.4: included samples comprised a lower proportion of seminomas (32%) and platinum sensitive/unselected tumours (71%) compared to those excluded from these analyses (54% and 83%, respectively).

**Copy number signature identification**. We sought evidence for the presence of somatic copy number signatures previously deliniated in ovarian cancer in TGCT using the approach described in ref. [27]. We utilised the genome-wide distribution of the following six different copy number features: (i) segment size, the bp length for each segment; (ii) breakpoint count per 10 MB, the number of breakpoints in 10 MB sliding windows across the genome; (iii) change-point copy number, the absolute difference between adjacent copy number segments; (iv) segment copy number, the absolute copy number of the segment; (v) breakpoint count per chromosome arm, the number of breaks per chromosome arm; (vi) length of segments with oscillating copy number, the number of contiguous copy number segments switching between two copy number states, rounded to nearest integer state. In ref. [27], the density distributions of these six genomic features were separated out into a total of 36 distinct components using mixture modelling on high quality absolute CN profiles from 91 whole genome sequenced ovarian cancer tumours: Gaussian mixture models were used for features i, iii and iv whilst Poisson mixture models were used for features ii, v and vi, with the number of components selected based on lowest Bayesian Information Criterion. We took these 36 distinct components, and for each copy number event identified in a TGCT sample, we computed the posterior probability of that event belonging to a component. For each individual TGCT sample, these posterior event vectors were summed resulting in a sum-of-posterior probabilities vector, which were combined in a patient-by-component sum-of-posterior probabilities matrix. The R package NMF (v0.21.0) was used to deconvolute the patient-by-component sum-of-posteriors matrix into a patient-by-signatures matrix and a signature-by-component matrix. Non-integer copy number values were used in all analyses, as some of the signatures are dependent on these values. Low purity samples (cellularity < 40%) were not included in these analyses as signature quantification in such samples is insufficiently reliable. Two-tailed Pearson's correlation was computed between the signature-by-component weight matrices between each of datasets using the cor. test() function using the R package stats (v3.5.1). These analyses were performed on copy number calls from TGCT samples from the TCGA series ($n = 105$) and the ICR1 series ($n = 19$) and compared with copy number calls from 415 SNP array profiling of ovarian cancer cases as part of TCGA as derived in ref. [27].

**Gene expression data**. RSEM gene expression data for the TCGA cohort was obtained via the firehose browser[51]. We used the R package edgeR (v3.24.3) to calculate log2 ratio reads per kilobase million (RPKM) values, filtering out lowly expressed genes (<2 reads per million in >90% of samples). Data were normalised using the calcNorm function, which normalises for RNA composition via scaling factors for library sizes that minimise the log-fold changes between samples for the majority of genes.

**CN signature exposure comparisons and correlations**. We compared CN signature exposure values between cases according to molecular features (presence/absence of mutation in KIT, presence/absence of mutation RAS pathway genes (KRAS, NRAS)), via Wilcoxon signed rank test using the wilcox.test() function implemented in the R package stats (v3.5.1). False discovery rate was controlled for via the Benjamini-Hochberg method. We analysed correlation of age, log2 ratio RPKM, chromosome/gene copies with CN signature exposures. Pearson correlation coefficients were calculated using the cor.test() function from the R package stats (v3.5.1).

**Force calling SNVs**. To compare presence/absence of mutations between different tumours from the same patient, we looked for evidence of a particular variant called in a particular tumour for its presence, even at low allelic fraction, in all other tumours from the same patient. The strong prior of having been detected de novo

in one tumour enables more sensitive detection in a second related tumour; a method termed force-calling[62]. We used the overlap of calls from MuTect, MuTect2 and Strelka as described above to generate an aggregate set of somatic SNVs for each patient. We then used SAMtools to count the number of reads supporting the reference and alternate alleles at those sites in their matched samples. Reads were considered only if they were from unique pairs, had a variant base quality of ≥20 and a read quality of ≥5. Two supporting bases were required to consider the variant present in a matched tumour, whilst less than two supporting bases at a minimum of 50× were required to consider the variant absent. Variants were classified as truncal if they were present in all tumour samples available from an individual, primary only if they were confined to the primary tumour and metastasis only if they were confined to the metastatic tumour(s) from a given case. Gene set enrichment analysis was performed on nonsynonymous mutations identified in each category using g:Profiler[63] (rev 1760) and adjusted for multiple tests using a g:SCS corrected threshold of $p < 0.05$.

**Phylogenetic trees**. Log2 ratio and B-allele frequency intensity data from the OncoSCan CNV Plus Assay were filtered so that only informative probes (i.e., those with a heterozygous genotype) at copy neutral regions within the matched normal sample were retained for analysis. We performed joint segmentation on tumour samples grouped by case using the asmultipcf function in the R package ASCAT (v2.1.1). Phylogenetic trees were constructed from major and minor copy number calls from ASCAT (capped at a maximum of 4n) using the nearest neighbour joining method in MEDICC[28,29]. Tree phylogenies were rooted to an assumed pure diploid outgroup with no copy number changes.

## Data availability

The ICR2 WES and MIP data files are available from EGA (accession: EGAS00001003811). The DFCI WES data files are available from dbGaP (phs000923.v1.p1). TCGA SNV (TGCT Mutation Packager Calls Level 3) and RSEM (TGCT RSEM genes normalized data Level 3) data are available via the firehose browser (https://gdac.broadinstitute.org/). TCGA SNP array data (Affymetrix SNP Array 6.0 CEL files) are available from the GDC Legacy Archive (https://portal.gdc.cancer.gov/legacy-archive/search/f). SNV calls from the MSK and FDM datasets and CN calls from the DFCI and ICR1 are available from their respective publications. The remaining data are within the supplementary files or are available from the authors upon reasonable request.

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

## Acknowledgements

We thank all the individuals who took part in these studies and all the researchers, clinicians, technicians and administrative staff who have enabled this work to be carried out. We acknowledge the National Health Service funding to the National Institute for Health Research Biomedical Research Centre. This study was supported by the Movember foundation and the Institute of Cancer Research. K. Litchfield is supported by a PhD fellowship from Cancer Research UK. R.S.H. and P.B. are supported by Cancer Research UK (C1298/A8362 Bobby Moore Fund for Cancer Research UK).

## Author contributions

Designed the study: C.T., K.L., C.L. Identification and recruitment of patients: R.H., T.P., S.C. Coordinated sample administration and tracking: B.B., D.D. Clinical parameter definition: E.G., M.A.B., C.H., C.S., R.H., A.R., C.T. Coordinated management of tumour tissue and DNA extraction: K.L., P.Z.P., M.L., F.S., D.G.C. Undertook histological examination of tumour tissue: S.O. Designed laboratory experiments: K.L., C.T. Conducted laboratory experiments: P.Z.B., A.H., P.B. Designed bioinformatics analyses: C.L., C.T. Development of bioinformatics algorithms: J.D.B., G.M., A.J.C. Performed bioinformatics and statistical analyses: C.L.

Drafted the manuscript: C.T., C.L., R.S.H., J.D.B., J.S. All authors reviewed and contributed to the manuscript.

## Competing interests

The authors declare no competing interests.
