## [Peer Review File · Nature Communications]

Reviewers' comments:

Reviewer #1 (Remarks to the Author):

The authors collated six public cohorts with DNA sequencing (WES and panel sequencing) from 631 patients. The goal of the work was to more clearly define the genomic alterations associated with platinum response.

With the focus being genomic alterations of resistant/sensitive tumors, figure 3 would be better served reorganized similar to figure 5 by platinum response, stage and histology instead of by gene.

Do you think the lower mutational frequency of TP53 and PTEN is due to the different sample sets with more resistant and metastases as well as extra-gonadal samples in the panels or is it that the variant allele frequencies is lower and only picked up by the higher coverage of targeted panels. VAF were not provided in Supplemental Table 7 so not fully clear though the differences in cohorts seems more likely.

The extra-gonadal samples were associated with almost every feature and I'm wondering if part of this due to the smaller numbers (n=50). KIT mutations are associated with platinum sensitivity and TP53 with platinum resistance, but both are associated with extra-gonadal disease.

For the 93 samples were removed from analysis with tumor purity <0.4 and was it biased by histology or platinum response? For Figure 4, was histology taken into account in the resistant tumors. Since there were few seminomas (supp table 3), were similar results in the resistant tumors when restricted to non-seminomas?

For Figure 5, while no single gene seemed to be associated with platinum response, the resistant samples do look overall more altered. Did you try calculating an overall score for CN alterations of those genes?

In the SNV mutational signatures, SNV-Sig-3 was discussed but seems that also Signature 1 was interesting as completely absent in the sensitive seminomas.

The copy number signatures section was really descriptive and quite long without focusing on the main point of this manuscript - sensitivity and resistance response. The analysis was only performed in unselected TGCTs (TCGA and ICR1) with a comment that it wasn't effective in the FFPE cohorts with formation about platinum response. I would strongly consider shorting or removing as it doesn't add to the story you are developing.

The number of 40 platinum-resistant TGCTs in the abstract and introduction is a little misleading as the results list that it is 40 samples from 26 cases (22 primaries and 18 metastases). This brings up another issue that it should be clearly documented in each figure the cohorts used and number of samples/cases for each analysis. While Supplemental Figure 1 was supposed to be the overview for this (and is a nice overview), the numbers are not consistent with the text and doesn't list which cohorts. Or maybe add to Supp Table 4 which samples were used in which analyses for clarity for repeatability of your analyses.

Supplemental tables could use additional information for clarity in understanding components/abbreviations.

I didn't seem to see any information about the public repository where the ICR2 sample data are deposited.

Minor comments:

1. What was the source of normal for the ICR2 set. The methods section only mentions tumor tissue.
2. In the methods, lines 511 states that the unselected tumors were grouped with the resistant tumors for analysis of platinum response. For all other analyses, the unselected were grouped with the sensitive.
3. Copy number analyses were corrected for age, but were mutations and tumor mutational burden associated with age as well?
4. Figure 3C, the color coding for stage and histology is different than the other panels. It might also be useful to add the % mutated (or number) next to the genes in a since only a small portion of the tumors are plotted. It might also be useful to add an annotation track with whether the samples was measured by WES or panel.
5. Would be beneficial to add in the legend for figure 5.
6. Supp Figure 4. For B, is the difference between TCGA and ICR1 due to subtype distribution? In C, what is the axes, proportion?
7. Supp Figure 5, what are the gray blocks? They are not defined in text. Why are some mutations highlighted?
8. Supp Figure 6. Could you add histology?

Reviewer #2 (Remarks to the Author):

Review of "Genomic landscape of platinum resistant and sensitive testicular cancers" by Loveday et al.

The study by Loveday and co-authors details the genomic landscape of platinum-sensitive and resistant testicular cancers. Since cisplatin and other platinum drugs are the mainstay of testis cancer therapy, the topic is of clinical importance. The authors determine somatic SNVs and copy-number changes from a set of clinically annotated tumors, supplemented by similar data sets from previously published analyses. A number of different platforms were used in these studies, including whole-exome and panel sequencing, molecular inversion probes and SNP arrays. Integrating the data, the authors claim to have identified 7 features of testicular tumours. Briefly, these are: 1) overall low tumour mutation burden, with higher TMB in platinum-resistant tumours; 2) prominence of KIT and RAS-pathway alterations, especially in seminomas, and association of KIT mutation with platinum sensitivity; 3) low frequency of somatic TP53 mutation; 4) WNT/CTNNB1 aberrations in resistant and metastatic GCTs; 5) widespread structural aberration, with an attempt to link specific copy-number signatures to a molecular cause; 6) evolution of CN aberration during disease progression; and 7) prominence of homologous recombination deficiency.

Overall, the study is well-done and clearly described. The figures are particularly good. The study addresses an important topic. Many of the seven hallmarks been identified in previous studies, and in some cases the authors make statements that are perhaps stronger than warranted by the evidence. Nonetheless, this is a valuable contribution that will help advance the field. There are several considerations that could strengthen the manuscript if addressed:

Specific comments:

1. Previous studies (e.g. Bagrodia et al, J Clin Oncol 2016) identified an association between p53 mutation/MDM2 overexpression and platinum resistance, a finding that was not duplicated in the present study. Since the JCO article used panel-based sequencing and this ms included other approaches such as MIPs and WES, can the authors please comment about the possible effects of

read-depth and the resolution of the different platforms used for CN determination?

2. The logistic regression model identified an association between KIT mutation and platinum sensitivity. However, if the data are considered as 2x2 contingency tables, there is no significant association (see reviewer attachment). Can the authors explain this apparent discrepancy? It seems more likely that the results show that KIT mutations are more common in seminomas, and seminomas are more likely than non-seminomas to be platinum-sensitive. This is an important point—it would almost certainly be a mistake to intensify therapy for a seminoma that was clinically genotyped and found to lack a KIT mutation, which is one possible logical clinical extension of the authors' claim.

3. What is the significance of CN-6 being associated with 12p amplification and KRAS amplification/overexpression? Wouldn't that be expected, since CN-6 reflects increased copy number from focal amplification?

4. In Figure 7c, it appears that CN-1 is significantly higher in seminomas with KIT mutation compared to seminomas without. Is the same true for RAS? The text states that it is so (line 210) but the figure does not show statistics. This observation (in the case of KIT, and possibly RAS) is very interesting and the authors are correct to emphasize it. However, the phrasing of the abstract ("These analyses provide the first mechanistic links between driver mutations in RAS and KIT and the widespread copy number events by which TGCT is characterized") I think somewhat overstates the case. What precisely is the mechanism being posited? What is the direct experimental evidence to support this claim? Again, because these mutations are most common in one subtype (seminoma), there is a risk of confounding the effect of histologic subtype with effects of driver mutations.

5. In the discussion can the authors please address the possible bias introduced by the inclusion of large numbers of platinum-resistant tumors in nearly all the datasets used (except TCGA)? While including these tumors is obviously critical to some of the questions explored, for the more general conclusions drawn in this study about GCT tumorigenesis, the caveat should be noted that the dataset is not completely representative of the clinical spectrum of GCTs (i.e. resistant tumors are overrepresented).

(1)Reviewer 1

The authors collated six public cohorts with DNA sequencing (WES and panel sequencing) from 631 patients. The goal of the work was to more clearly define the genomic alterations associated with platinum response.
With the focus being genomic alterations of resistant/sensitive tumors, figure 3 would be better served reorganized similar to figure 5 by platinum response, stage and histology instead of by gene.

Thank you for this suggestion. There is an inevitable dilemma when preparing an 'oncoplot' annotated by a number of parameters as to which parameter to prioritise. In Figure 3a, we were seeking primarily to present the frequency of mutations by type (and how they overlapped); annotations regarding platinum response and other clinical parameters were thus necessarily secondary. Hence, we provided Figure 3b to demonstrate association at gene level with clinical parameters, showing directly by clinical parameter the proportions of mutated/non-mutated tumours (with p-values for significance).

Do you think the lower mutational frequency of TP53 and PTEN is due to the different sample sets with more resistant and metastases as well as extra-gonadal samples in the panels or is it that the variant allele frequencies is lower and only picked up by the higher coverage of targeted panels. VAF were not provided in Supplemental Table 7 so not fully clear though the differences in cohorts seems more likely.

Thank you for flagging this. The vagaries of varying clinical make-up of sample series underpin many of the conflicting findings reported in the literature. It is for exactly these reasons that we undertook adjusted analyses using multifactorial logistic regression. This allowed simultaneous adjustment by the various relevant clinical variables, e.g. histology, site (gonadal, extragonadal), platinum response, sequencing platform etc., which might otherwise confound observed associations.

As the reviewer highlights, there was indeed a marked over-representation in the MSK dataset of extragonadal tumours (Sup Table 4), a parameter well recognised as strongly associated with platinum resistance of site.

Once adjusted for site (gonadal vs extra-gonadal) and histology (non-seminomas are more likely to be platinum resistant), the association of TP53 mutation with platinum resistance transpired to be much weaker ($p=0.034$) than previously reported in Bagrodia et al 2016.

In our logistic regression, adjusted for all other clinical variables, we found association of TP53 mutation with platform (panel vs exome, Sup Table 10), as summarised in the table below:

Table of TP53 mutation counts:

	WES	Panel
Primary testicular tumours	1/254	1/148
Primary extragonadal	1/2	20/40
Metastatic	0/34	19/186

Like the reviewer, we speculated as to the possible explanations for this.

Like the reviewer, we speculated this might be a function of differential detection of subclonal TP53 mutations through the additional depth afforded by panel sequencing.

However, the VAF of TP53 mutations detected by panel sequencing was relatively high (median VAF = 0.44, IQR = 0.64-0.29) and in keeping with TP53 VAF detected by WES (VAFs = 0.13 and 0.96), thus providing no support for this hypothesis.

We were particularly keen to scrutinise adequacy of depth of coverage for the treatment resistant metastatic tumours analysed by WES as the absence of TP53 mutations detected in this subset of WES samples was particularly noteworthy (0/34 mets in WES samples vs 19/186 mets in panel samples). Analysis of the 34 treatment resistant metastatic tumours showed that 92% and 90% of bases in the TP53 CDS were covered at 20x or greater in tumour and normal samples, respectively. Importantly, coverage was similarly good across the TP53 DNA binding domain (94% and 89%, respectively).

Thus, panel sequencing is associated with TP53 mutation status for reasons that we cannot currently account for.

The extra-gonadal samples were associated with almost every feature and I'm wondering if part of this due to the smaller numbers (n=50). KIT mutations are associated with platinum sensitivity and TP53 with platinum resistance, but both are associated with extra-gonadal disease.

The use of logistic regression to adjust simultaneously for a number of parameters:

(i) accounts for the role of site (gonadal vs extragonadal) in the observed association of each of

(a) KIT mutation with platinum sensitivity

(b) TP53 with platinum resistance

(ii) takes account of the numbers in each 'cell' and degrees of freedom of the test in assigning significance to an observation.

Hence, the p-values of association of extragonadal disease with each clinical parameter are a reflection of the strength of association given the sample size analysed.

For the 93 samples were removed from analysis with tumor purity <0.4 and was it biased by histology or platinum response?

Thank you for highlighting this. Comparison of the excluded samples to the included samples demonstrates modest variation in both histology (seminomas comprised 54% vs 32%, respectively) and platinum response (sensitive/unselected comprised 83% vs 71% respectively). We have updated the methods section "Analyses for aneuploidy" to include this information.

"These analyses were performed in a subset of 188 tumours with purity estimates ≥ 0.4 : included samples comprised a lower proportion of seminomas (32%) and platinum sensitive/unselected tumours (71%) compared to those excluded from these analyses (54% and 83%, respectively)."

For Figure 4, was histology taken into account in the resistant tumors.

Since there were few seminomas (supp table 3), were similar results in the resistant tumors when restricted to non-seminomas?

The plots in Figure 4 are shown independent of histology. However, we have now updated Supplementary Tables 13 and 14 to include counts of events (and associated p values) by histology. These data show that the results are broadly similar when restricting the analyses to nonseminomas only.

For Figure 5, while no single gene seemed to be associated with platinum response, the resistant samples do look overall more altered. Did you try calculating an overall score for CN alterations of those genes?

Thank you for this suggestion. We have extended our analyses to generate a categorical variable for each of a) copy gain for TGCT oncogenes (≥ 5 copies equals copy gain vs < 5 copies) and b) copy loss for TGCT TSGs (≤ 1 copies equals copy loss vs > 1 copy), with the respective 2x2 contingency tables shown below. This allowed us to undertake multivariate adjusted analysis for gain and for loss, respectively, to examine for association by gene and (and gene group) for each clinical variable.

As illustrated below and in Supplementary Table 15 (new) there were no associations with platinum response at $p < 0.05$ after adjusting for histology, site and sample series.

a) All TGCT oncogenes (excl. *KRAS* which shows near near universal copy gain)

2x2 contingency table

	Resistant	Sensitive/unselected
No copy gain (<5 copies)	19	79
Copy gain (≥ 5 copies)	17	55

Logistic regression model

Deviance Residuals:

Min	1Q	Median	3Q	Max
-1.9066	-0.8844	-0.8548	0.9542	1.8991

Coefficients:

	Estimate	Std. Error	z value	Pr(> z)
(Intercept)	0.2913	0.8034	0.363	0.716891
Histologyseminoma	1.3490	0.3528	3.824	0.000131 ***
Platinum_Response_Category sensitive/unselected	-0.9528	0.9264	-1.028	0.303731
Sample_Series CR1	-0.4963	0.8721	-0.569	0.569274
Sample_Series CR2	-0.6444	0.7322	-0.880	0.378841
Sample_Series TCGA	0.3900	0.7152	0.545	0.585587
Stageprimary	-0.4655	0.8755	-0.532	0.594940

Signif. codes: 0 '***' 0.001 '**' 0.01 '*' 0.05 '.' 0.1 ' ' 1

(Dispersion parameter for binomial family taken to be 1)

Null deviance: 231.68 on 169 degrees of freedom
Residual deviance: 212.50 on 163 degrees of freedom
AIC: 226.5

Number of Fisher Scoring iterations: 4

b) all TGCT TSGs

2x2 contingency table

	resistant	sensitive/unselected
No copy loss (>1 copy)	19	107
Copy loss (≤ 1 copy)	17	27

Logistic regression model

Deviance Residuals:

Min	1Q	Median	3Q	Max
-1.7509	-0.5809	-0.3878	0.2624	2.3637

Coefficients:

	Estimate	Std. Error	z value	Pr(> z)
(Intercept)	-6.2014	1.8431	-3.365	0.000766 ***
Histologyseminoma	2.0618	0.4746	4.344	1.4e-05 ***
Platinum_Response_Category-sensitive/unselected	-0.1123	1.4085	-0.080	0.936438
Sample_Series1 CR1	1.0366	1.0169	1.019	0.308032
Sample_Series1 CR2	3.9077	1.2721	3.072	0.002127 **
Sample_Series1 TCGA	0.1807	0.8889	0.203	0.838925
Stageprimary	3.5833	1.4793	2.422	0.015422 *

Signif. codes: 0 '***' 0.001 '**' 0.01 '*' 0.05 '.' 0.1 ' ' 1

(Dispersion parameter for binomial family taken to be 1)

Null deviance: 194.42 on 169 degrees of freedom
Residual deviance: 137.50 on 163 degrees of freedom
AIC: 151.5

Number of Fisher Scoring iterations: 5

In the SNV mutational signatures, SNV-Sig-3 was discussed but seems that also Signature 1 was interesting as completely absent in the sensitive seminomas.

Thank you for highlighting this. For brevity we had restricted our observations regarding SNV signatures to SNV-Sig-3, but agree this observation is warranting of inclusion.

We have added the following text to the manuscript:

“Seminomas typically show global hypomethylation whilst non-seminomas exhibit increased methylation associated with increased differentiation, from embryonal carcinoma through to teratoma. SNV-Sig-1 is thought to reflect spontaneous deamination of 5-methylcytosine. The complete absence of SNV-Sig1 observed in platinum sensitive seminomas is thus in fitting with existing hypotheses of association in cancer between differentiation, global DNA methylation status and response to chemotherapy^{25,26}.”

The copy number signatures section was really descriptive and quite long without focusing on the main point of this manuscript - sensitivity and resistance response. The analysis was only performed in unselected TGCTs (TCGA and ICR1) with a comment that it wasn't effective in the FFPE cohorts with formation about platinum response. I would strongly consider shorting or removing as it doesn't add to the story you are developing.

TGCTs are genomically most characterized by structural aberration and disordered genomes, features that have previously only been described at high level and largely without correlation to other clinical or molecular parameters. Hence, our CN analyses of the unselected/sensitive fresh frozen series were novel, albeit not focused on platinum response. We undertook CN-signature analysis in the FFPE-derived resistant samples; unfortunately (but unsurprisingly), the data were deemed of insufficient quality to be included in the manuscript.

As Reviewer 2 highlights, the themes explored in these analyses relate not only to platinum resistance but address a broader enquiry of GCT tumorigenesis: and as Reviewer 2 notes, given this is the first presentation of copy number signature analysis in

a disease characterised by CNs, we feel it is a strength and novelty of the work.
The number of 40 platinum-resistant TGCTs in the abstract and introduction is a little misleading as the results list that it is 40 samples from 26 cases (22 primaries and 18 metastases). This brings up another issue that it should be clearly documented in each figure the cohorts used and number of samples/cases for each analysis. While Supplemental Figure 1 was supposed to be the overview for this (and is a nice overview), the numbers are not consistent with the text and doesn't list which cohorts. Or maybe add to Supp Table 4 which samples were used in which analyses for clarity for repeatability of your analyses.
We thank Reviewer 1 for these helpful suggestions regarding clarity in data tables and have updated the abstract, introduction and Supp Table 4 as recommended. Abstract: "40 tumours from 26 cases with platinum-resistant TGCT" Introduction "Here we present whole exome-sequencing (WES) on 40 tumours from 26 cases with platinum-resistant TGCT"
Supplemental tables could use additional information for clarity in understanding components/abbreviations.
We thank Reviewer 1 for these helpful suggestions regarding supplementary tables and have thoroughly updated all supplementary tables in regard of abbreviations contained therein.
I didn't seem to see any information about the public repository where the ICR2 sample data are deposited.
We thank Reviewer 1 for highlighting this: we have now updated the manuscript to include a data availability statement. "Data availability The ICR2 WES and MIP data files are available from EGA (accession: EGAS00001003811). The DFCI WES data files are available from dbGaP (phs000923.v1.p1). TCGA SNV and RSEM data are available via the firehose browser (https://gdac.broadinstitute.org/). TCGA SNP array data are available from the GDC Legacy Archive (https://portal.gdc.cancer.gov/legacy-archive/search/f). SNV calls from the MSK and FDM datasets and CN calls from the DFCI and ICR1 are available from their respective publications. The remaining data are within the supplementary files or are available from the authors upon reasonable request."
Minor Comments 1. What was the source of normal for the ICR2 set. The methods section only mentions tumor tissue.
Thank you for highlighting this omission. The source of normal for the ICR2 set was lymphocyte-derived DNA extracted using standard techniques. We have updated the first paragraph within the methods section to clarify the use of lymphocyte-derived DNA (rather than non-cancer parenchymal tissue): "Matched normal DNA was obtained from lymphocytes using standard techniques."
2. In the methods, lines 511 states that the unselected tumors were grouped with the resistant tumors for analysis of platinum response. For all other analyses, the unselected were grouped with the sensitive.
We are most appreciative of Reviewer 1 for spotting this typographical error. Indeed, the unselected tumours are grouped with sensitive tumours throughout, as unselected series

will comprise >90% platinum-sensitive samples. We have updated the text to reflect this.

3. Copy number analyses were corrected for age, but were mutations and tumor mutational burden associated with age as well?

We thank Reviewer 1 for raising this point.

Age at diagnosis was one of a number of parameters for which we examined for association with CN signatures (rather than correcting/adjusting for age). Age at diagnosis was not available for all of the sample series used in this study: hence we could not include it in the linear regression model for TMB.

We have now completed an analysis by age for TMB equivalent to that performed for copy number signatures to examine for association between TMB and age at diagnosis in the TCGA dataset. As anticipated, a positive correlation was observed, more prominent in nonseminomas (cor = 0.43, p-value = 0.0002), than seminomas (cor = 0.17, p-value = 0.17). This data is in keeping with the observation of the age-related COSMIC SNV signature one being predominantly present in nonseminomas. We have added the following text to the manuscript:

“TMB was positively correlated with age at diagnosis, more prominently in nonseminomas (cor = 0.43, p-value = 0.0002), than seminomas (cor = 0.17, p-value = 0.17).”

We have also updated the methods section to highlight why subgroup analysis rather than adjusted analysis was undertaken:

“Age at diagnosis, a potential cofounder, was not included in the regression model due to substantial missingness across the datasets.”

4. Figure 3C, the color coding for stage and histology is different than the other panels. It might also be useful to add the % mutated (or number) next to the genes in a since only a small portion of the tumors are plotted. It might also be useful to add an annotation track with whether the samples was measured by WES or panel.

We are most appreciative of Reviewer 1 for spotting this inconsistency. We have now updated the colour scheme for full consistency and ease of readership. We have also added % mutated and an annotation track for WES vs panel to the figure.

5. Would be beneficial to add in the legend for figure 5.

We are most appreciative of Reviewer 1 for highlighting this. We have now added a full legend for Figure 5.

6. Supp Figure 4. For B, is the difference between TCGA and ICR1 due to subtype distribution? In C, what is the axes, proportion?

Thank you; an interesting question. The pattern of CN-signatures and correlations with histology and other clinical correlates are overall consistent between TCGA and ICR1. These are both FF series which comprise predominantly sensitive samples; the histology is overall comparable (% seminomas/nonseminomas: TCGA 43/57, ICR1 52/48). Thus the differences in the constituent sub-features underpinning the CN signatures elucidated via Supp. Fig 4 may in part reflect difference at series level in constituency, but likely are more driven by platform difference (array versus WES-derived CN calls).

We have now updated Supp. Figure 4C (and Figure 7C) to include a title for the y axis: “Signature Exposure Level”.

7. Supp Figure 5, what are the gray blocks? They are not defined in text. Why are some mutations highlighted?
Thank you. We are most appreciative of Reviewer 1 for highlighting this. We have expanded the legend for Supp. Figure 5 to clarify that the grey bars reflect synonymous mutations. The mutations highlighted in the schematic comprise those in TGCT driver genes and genes that were significant in the gene set enrichment analysis (this has been added to Figure legend).
8. Supp Figure 6. Could you add histology?
Thank you for this suggestion. We have updated Supp. Figure 6 to include details of the histology of samples for which we have presented phylogenetic data.
Reviewer 2
The study by Loveday and co-authors details the genomic landscape of platinum-sensitive and resistant testicular cancers. Since cisplatin and other platinum drugs are the mainstay of testis cancer therapy, the topic is of clinical importance. The authors determine somatic SNVs and copy-number changes from a set of clinically annotated tumors, supplemented by similar data sets from previously published analyses. A number of different platforms were used in these studies, including whole-exome and panel sequencing, molecular inversion probes and SNP arrays. Integrating the data, the authors claim to have identified 7 features of testicular tumours. Briefly, these are: 1) overall low tumour mutation burden, with higher TMB in platinum-resistant tumours; 2) prominence of KIT and RAS-pathway alterations, especially in seminomas, and association of KIT mutation with platinum sensitivity; 3) low frequency of somatic TP53 mutation; 4) WNT/CTNNB1 aberrations in resistant and metastatic Overall, the study is well-done and clearly described. The figures are particularly good. The study addresses an important topic. Many of the seven hallmarks been identified in previous studies, and in some cases the authors make statements that are perhaps stronger than warranted by the evidence. Nonetheless, this is a valuable contribution that will help advance the field. There are several considerations that could strengthen the manuscript if addressed:
We concur with Reviewer 2 that improving insights into the molecular correlates of platinum response is of high clinical importance and thank him/her for the comments regarding the study being “well done”, “clearly described”, “ a valuable contribution” and “advancing the field” as well as the figures being “particularly good”. We thank the reviewer for providing the most helpful constructive criticism we address hereafter, by which we have sought, as (s)he suggests, to “strengthen the manuscript”.
1. Previous studies (e.g. Bagrodia et al, J Clin Oncol 2016) identified an association between p53 mutation/MDM2 overexpression and platinum resistance, a finding that was not duplicated in the present study. Since the JCO article used panel-based sequencing and this ms included other approaches such as MIPs and WES, can the authors please comment about the possible effects of read-depth and the resolution of the different platforms used for CN determination?
Through multifactorial adjusted analysis using logistic regression (controlling for histology, site, stage, treatment response, sequencing platform and study), we are able to control for differences between sample series in terms of clinical parameters. Extragonadal tumours were substantially over-represented in the MSK series reported by Bagrodia et al. (2016), as were nonseminomas. Both extragonadal site and nonseminomatous pathology are associated with poorer outcome (i.e. platinum resistance). Hence, following adjustment for site and histology, the residual association of TP53 with platinum response is marginal

(Supp. Table 10, Fig 3).

We also found a substantial rate of MDM2 copy gain across the TGCT series (see Figure 5).

To examine explicitly clinical associations of amplification in particular genes, we have extended our analyses to generate a categorical variable for a) copy gain for TGCT oncogenes (≥ 5 copies equals copy gain vs < 5 copies) and b) copy loss for TGCT TSGs (≤ 1 copies equals copy loss vs > 1 copy). This allowed us to undertake multivariate adjusted analysis for amplification and for loss, to examine for association by gene and globally with each clinical variable of amplification or loss respectively.

Through this multifactorial adjusted analysis, we did not find an association of MDM2 copy gain with platinum resistance ($P = 0.81$).

2. The logistic regression model identified an association between KIT mutation and platinum sensitivity. However, if the data are considered as 2x2 contingency tables, there is no significant association (see reviewer attachment). Can the authors explain this apparent discrepancy? It seems more likely that the results show that KIT mutations are more common in seminomas, and seminomas are more likely than non-seminomas to be platinum-sensitive. This is an important point—it would almost certainly be a mistake to intensify therapy for a seminoma that was clinically genotyped and found to lack a KIT mutation, which is one possible logical clinical extension of the authors' claim.

There is a significant association of KIT mutation with platinum sensitivity ($P = 7.84 \times 10^{-4}$) which is independent in multivariate analysis of the association of KIT mutation with histology ($P = 3.3 \times 10^{-10}$) and with extra gonadal disease ($P = 1.72 \times 10^{-3}$) (Supp. Table 10). We reconstruct the 2x2 contingency tables below to demonstrate that KIT mutation status IS associated with platinum response in a univariate analysis.

All tumours			
	Sensitive	Resistant	Totals
KIT mutant	45	10	55
KIT wildtype	230	233	463
Totals	275	243	518
Fisher exact < 0.0001			

Seminomas only			
	Sensitive	Resistant	Totals
KIT mutant	42	6	48
KIT wildtype	81	54	135
Totals	123	60	183
Fisher exact < 0.0003			

Non-seminomas only			
	Sensitive	Resistant	Totals
KIT mutant	3	4	7
KIT wildtype	149	179	328
Totals	152	183	335
Fisher exact = 1			

All tumours	Sensitive	Resistant	Totals
Seminom	123	60	183
Nonseminom	152	183	335
Totals	275	243	518

Fisher exact < 0.0001

3. What is the significance of CN-6 being associated with 12p amplification and KRAS amplification/overexpression? Wouldn't that be expected, since CN-6 reflects increased copy number from focal amplification?

In our presentation of the CNV signature analysis, we sought to demonstrate that particular hallmark features of TGCT are constituted by their expectant signatures. We agree with the reviewer that the association of KRAS/12p copy number with CN-sig-6 was an expected finding due to the nature of the signature itself.

4. In Figure 7c, it appears that CN-1 is significantly higher in seminomas with KIT mutation compared to seminomas without. Is the same true for RAS? The text states that is it so (line 210) but the figure does not show statistics. This observation (in the case of KIT, and possibly RAS) is very interesting and the authors are correct to emphasize it. However, the phrasing of the abstract ("These analyses provide the first mechanistic links between driver mutations in RAS and KIT and the widespread copy number events by which TGCT is characterized") I think somewhat overstates the case. What precisely is the mechanism being posited? What is the direct experimental evidence to support this claim? Again, because these mutations are most common in one subtype (seminoma), there is a risk of confounding the effect of histologic subtype with effects of driver mutations.

We agree with Reviewer 2 that the association of KIT and RAS mutation with CN-sig 1 are very interesting, in particular since this recapitulates an observation reported in a distinct tumour type also characterised by a highly disordered genome (serous ovarian cancer).

We have rephrased to clarify the individual associations for seminomas with *KIT* mutations ($P=0.004$) and *RAS* mutations (*KRAS* & *NRAS*; $P=0.07$) with the following text:

"...was the predominant signature and was higher in seminomas positive for mutations in *KIT* ($p = 0.004$) and *RAS* mutations (*KRAS* and *NRAS* combined, $p = 0.07$) ..."

We had sought not to over-egg the significance of this observation, nor leap to an inference of causality. As reviewer 2 highlights, these are observational data and assertion of causality would require time-course experimental data. We have modified our statement accordingly to:

"Albeit preliminary and observational in nature, these analyses provide the first support for a possible mechanistic link between early driver mutations in *RAS* and *KIT* and the widespread copy number events by which TGCT is characterised."

We did not seek to speculate unduly, but had included some preliminary hypotheses regarding potential underlying mechanism:

"Aberrant *RAS* signalling promotes genomic instability due to aberrant cell cycle checkpoint control leading to chromosomal missegregation^{33,34}. In HGSOC, CN-Sig-1 is directly correlated with telomere shortening and frequency of amplification-associated fold-back inversion events, pointing to the underlying mechanism as breakage-fusion-bridge²⁵."

5. In the discussion can the authors please address the possible bias introduced by the inclusion of large numbers of platinum-resistant tumors in nearly all the datasets used (except TCGA)? While including these tumors is obviously critical to some of the questions explored, for the more general conclusions drawn in this study about GCT

tumorigenesis, the caveat should be noted that the dataset is not completely representative of the clinical spectrum of GCTs (i.e. resistant tumors are overrepresented).

Thank you for highlighting this. The vagaries of varying clinical make-up of sample series underpins many of the conflicting findings reported in the literature. It is for exactly these reasons that we undertook adjusted analyses using multivariable logistic regression. This allowed simultaneous adjustment by the various relevant clinical variables, e.g. histology, site (gonadal, extra-gonadal), platinum response, sequencing platform etc., which might otherwise confound observed associations. This approach enables us to derive valid observations of association whilst also benefiting from additional power afforded by oversampling of rarer clinical groups (e.g. resistant and extragonadal samples).

REVIEWERS' COMMENTS:

Reviewer #1 (Remarks to the Author):

The authors have mostly addressed my comments. I still think the copy number signatures section could be tightened. In addition, it might be good to discuss the TP53 mutation status association with panel sequencing being unknown might be good to add to the discussion.

Reviewer #2 (Remarks to the Author):

The authors have satisfactorily addressed the concerns raised in the first round of reviews. I believe the manuscript is now suitable for publication.

Reviewer #3 (Remarks to the Author):

As a statistician, I think the reviewer 1 and 2 both asked a very good question on multifactorial logistical regression analysis, for which many published papers wrongly used the concept. A repeatedly used example that I used in the course is that, while people like to wear T-shirt in summer time, people also prefer to eat ice cream in summer time. If we use multifactorial logistical regression analysis in some, but not all, real-world datasets, we will see significant P value for a trend that people who like to wear T-shirt prefer to eat ice cream, independent of summer time. But we know that wearing T-shirt is not the cause of eating ice cream. In this situation, these significant P value might not be useful to determine cause-and-effect relationship. Similarly, in this manuscript, the authors suggested a significant statistical association between KIT mutation and platinum sensitivity, independent of histology, through multifactorial logistical regression analysis. After reviewing the details of method part without getting hands on the real data, it is possible that these significant P values can be generated from multifactorial logistical regression analysis. However, without real molecular experiments to prove that KIT mutation is the cause for platinum sensitivity, we have to be very careful if we try to use multifactorial logistical regression to understand cause-and-effect relationships. It is possible to get misled by multiple regression analysis results, and we should always use the results more as a suggestion, rather than for hypothesis testing.

So my suggestion is that the authors need to tune down their conclusion on this part, and have some sentences to discuss this issue. Most importantly, as Reviewer 1 mentioned the possible clinical application for KIT mutation in GCT patients, the authors should not leave the readers with impression that there is a causative association between KIT mutation and platinum sensitivity without experimental evidence.

Reviewer #1 (Remarks to the Author)
The authors have mostly addressed my comments.
We thank the reviewer for their updated comments and are pleased the reviewer is largely satisfied.
I still think the copy number signatures section could be tightened.
We have tightened up this section but only minorly. Publication on analysis of copy number signatures are still infrequent and we consider it instructive to the community to give reasonably comprehensive detail in regard of methods, findings and inferences. Furthermore, TGCT is characterised predominantly by aberration of copy number and exposition of the links between these signatures and the relevant molecular and aetiological factors would seem highly apposite to the overall intent of the landscape endeavour.
In addition, it might be good to discuss the TP53 mutation status association with panel sequencing being unknown might be good to add to the discussion.
We have added the following text discussing the association between TP53 mutation status and panel sequencing to the discussion: “We identified association between TP53 mutation and panel sequencing methodology, residual after adjustment for potentially confounding parameters. TP53 variant allele frequencies were typically high, and coverage of TP53 was sufficiently adequate in the exomes, to rule out differential detection of subclonal TP53 mutations between the two methods. It is possible the observed association may be due to confounding factors unaccounted for in these analyses, for example systematic differences in the clinical make-up of the constituent sample series.”
Reviewer #2 (Remarks to the Author):
The authors have satisfactorily addressed the concerns raised in the first round of reviews. I believe the manuscript is now suitable for publication.
Thank you. We appreciate your review and this conclusions
Reviewer #3 (Remarks to the Author):
As a statistician, I think the reviewer 1 and 2 both asked a very good question on multifactorial logistical regression analysis, for which many published papers wrongly used the concept. A repeatedly used example that I used in the course is that, while people like to wear T-shirt in summer time, people also prefer to eat ice cream in summer time. If we use multifactorial logistical regression analysis in some, but not all, real-world datasets, we will see significant P value for a trend that people who like to wear T-shirt prefer to eat ice cream, independent of summer time. But we know that

wearing T-shirt is not the cause of eating ice cream. In this situation, these significant P value might not be useful to determine cause-and-effect relationship. Similarly, in this manuscript, the authors suggested a significant statistical association between KIT mutation and platinum sensitivity, independent of histology, through multifactorial logistical regression analysis.

After reviewing the details of method part without getting hands on the real data, it is possible that these significant P values can be generated from multifactorial logistical regression analysis. However, without real molecular experiments to prove that KIT mutation is the cause for platinum sensitivity, we have to be very careful if we try to use multifactorial logistical regression to understand cause-and-effect relationships. It is possible to get misled by multiple regression analysis results, and we should always use the results more as a suggestion, rather than for hypothesis testing.

So my suggestion is that the authors need to tune down their conclusion on this part, and have some sentences to discuss this issue. Most importantly, as Reviewer 1 mentioned the possible clinical application for KIT mutation in GCT patients, the authors should not leave the readers with impression that there is a causative association between KIT mutation and platinum sensitivity without experimental

We thank the reviewer for their thoughtful comments regarding this part of our manuscript. We wholly agree with the reviewer that our analyses demonstrating association between *KIT* mutations and platinum sensitivity is (i) only as strong as the p-value indicates (ii) in no way infers a causal relationship and like any association could alternatively be due to chance, bias, confounding or reverse causality. Accordingly, whilst we have highlighted the association but not have sought to assert conclusions regarding causality. We agree that clinical inference of *KIT* mutation as a biomarker of platinum sensitivity would be inappropriate and have sought through the below addition to assert that our insights are immature with regard to the causal mechanisms underpinning these data.

“The significant associations observed may reflect causal relationship but may also arise due to the influences of chance, bias, confounding or reverse causality.